# TOWARDS CONTINUOUS MACHINE LEARNING ON PERIODIC CRYSTALS BY ULTRA-FAST INVARIANTS

## ABSTRACT

Periodic point sets model all solid crystalline materials (crystals) whose atoms can be considered zero-sized points with or without atomic types. This paper addresses the fundamental problem of checking whether claimed crystals are novel, not noisy perturbations of known materials obtained by unrealistic atomic replacements. Such near-duplicates have already skewed ground truth because past comparisons relied on discontinuous cells and symmetries. The proposed Lipschitz continuity under noise is a new essential requirement for machine learning on any data objects that have ambiguous representations and live in continuous spaces. For periodic point sets under isometry (any distance-preserving transformation), we designed the invariants that distinguish all known counter-examples to the completeness of past descriptors and detect thousands of (near-)duplicates in the world's five largest databases in a few minutes on a modest desktop computer.

## 1 MOTIVATIONS FOR CONTINUOUS INVARIANTS OF PERIODIC CRYSTALS

Real data such as periodic crystals often have ambiguous representations in the sense that experimental databases contain many substantially different entries encoding near-duplicate crystals with essentially the same properties Peplow (2023). Using descriptors or representations that discontinuously change under tiny perturbations of input data can lead to unjustified claims Krämer (2021) and even to 'paper mills' Bimler (2022) reporting thousands of 'new' materials without proof. These public investigations Francis (2023) already led to hundreds of retracted papers Chawla (2022). Machine learning can avoid such embarrassment by embracing a new *continuous* approach to data.

Any discovery should be validated by proper measurements, which are formalized by the concept of a distance *metric $d$* satisfying three axioms. The first axiom says that the distance $d(S, Q) = 0$ between any materials $S, Q$ vanishes if and only if $S, Q$ are the same. What materials should be called 'the same' Sacchi et al. (2020)? The relation of being 'the same' is called an *equivalence $S \sim Q$* if the following axioms hold: (1) any object is equivalent to itself $S \sim S$, (2) *symmetry*: if $S \sim Q$ then $Q \sim S$, (3) *transitivity*: if $S \sim Q$ and $Q \sim T$ then $S \sim T$. The transitivity axiom is especially important by justifying a classification into disjoint *equivalence classes* $[S] = \{Q \mid Q \sim S\}$. If two such classes $[S]$ and $[T]$ share a common object $Q$, they should coincide by transitivity: $[S] = [T]$.

A scientific approach is to first define an equivalence and then look for properties that can distinguish non-equivalent objects. For example, all crystals form disjoint classes by their chemical composition, though diamond and graphite composed of pure carbon have vastly different properties.

Because crystal structures are determined in a rigid form, the strongest equivalence (best separating all crystals) is *rigid motion*, which is a composition of translations and rotations in $\mathbb{R}^n$ from the group $\mathrm{SE}(n)$. Because noise perturbs any rigid structure, all $\mathrm{SE}(n)$-classes of crystals form a *continuous* space. The slightly weaker *isometry* (denoted by $S \simeq Q$) is defined as any distance-preserving transformation or, equivalently in $\mathbb{R}^n$, any composition of a rigid motion and a reflection. All isometries of $\mathbb{R}^n$ form the Euclidean group $\mathrm{E}(n)$. A classification under isometry suffices in practice because any mirror images can be distinguished by an extra bit (a sign of orientation).

**Definition 1.1** (periodic point set $S$ with a motif $M$). *Any basis of vectors $v_1, \ldots, v_n$ in $\mathbb{R}^n$ defines the* lattice $\Lambda = \{\sum_{i=1}^{n} c_i v_i \mid c_i \in \mathbb{Z}\}$ *and* unit cell $U = \{\sum_{i=1}^{n} t_i v_i \mid 0 \le t_i < 1\}$. *For a finite set $M \subset U$ (called a* motif*), the* periodic point set *is $S = M + \Lambda = \{p + v \mid p \in M, v \in \Lambda\} \subset \mathbb{R}^n$.* ∎

Figure 1: **Left**: any periodic point set can be given by many pairs (cell, motif), see Definition 1.1. Finite subsets of the same lattice within boxes or balls of the same cut-off size can be vastly different. **Right**: almost any perturbation of atoms can arbitrarily scale up a unit cell and break the symmetry.

A traditional representation of crystals is ambiguous in the sense that infinitely many pairs (cell, motif) generate the same periodic set of points, see Fig. 1 (left). Much worse, Fig. 1 (right) shows that any cell-based representation is inherently discontinuous because almost any perturbation of atoms (due to ever-present thermal vibrations and measurement noise) can arbitrarily scale up a *minimal* (by volume) cell. The past approach to ignore atomic perturbations up to a threshold $\varepsilon$ implies that sufficiently many tiny perturbations can transform any infinite set of points into any other Zwart et al. (2008), which makes *all sets equivalent* by the transitivity axiom. Tiny differences between crystals should be not ignored but quantified by a *continuous* metric as formalized below.

**Problem 1.2.** *Find a descriptor $I$ of all periodic point sets in $\mathbb{R}^n$ satisfying the conditions below.*

*(a) **Invariance**: if $S \simeq Q$ are isometric then $I(S) = I(Q)$, i.e. $I$ has no false negatives.*

*(b) **Completeness**: for any $S, Q \subset \mathbb{R}^n$, if $I(S) = I(Q)$ then $S \simeq Q$, i.e. $I$ has no false positives.*

*(c) **Metric axioms**: there is a distance metrics $d$ on invariant values satisfying three axioms (1) $d(a, b) = 0$ if and only if $a = b$, (2) $d(a, b) = d(b, c)$, (3) $d(a, b) + d(b, c) \geq d(a, c)$ for all $a, b, c$.*

*(d) **Continuity**: if $Q$ is obtained by perturbing every point of $S$ up to $\varepsilon$, then $d(I(S), I(Q)) \leq \lambda \varepsilon$.*

*(e) **Reconstructability**: any periodic set $S$ can be reconstructed from $I(S)$ up to isometry of $\mathbb{R}^n$.*

*(f) **Computability**: for a fixed dimension $n$, the invariant $I(S)$, the metric $d$ in (c), and a reconstruction of $S \subset \mathbb{R}^n$ from $I(S)$ in (e) are computable in polynomial time of the motif sizes.* ∎

The invariant $I(S)$ can be a vector, matrix, or another object in a space where metric computations should be easier than for isometry classes of $S$. Invariance condition 1.2(a) is stronger than *equivariance* saying that a group action $f$ (such as a rotation) changes $I(S)$ to $T_f(I(S))$, where $T_f$ is a map depending on $f$. For example, any linear combination $e(S)$ of point coordinates of $S$ is equivariant but can allow a false negative that is a pair $S \simeq Q$ with $e(S) \neq e(Q)$. The invariance means that $T_f$ is the identity, hence different values $I(S) \neq I(Q)$ always guarantee that $S \not\simeq Q$ are not isometric.

Completeness 1.2(b) is harder and is *practically meaningful only with a Lipschitz continuous metric* in 1.2(d) because any noise makes all real objects at least slightly different as in Fig. 1 (right). This unresolved discontinuity created a gigantic loophole that allows anyone to *disguise known materials as new* by perturbing atomic positions, which scales up a minimal cell, and by changing atomic types, which makes comparisons by symmetries, unit cells, and chemical compositions unreliable.

The metric axioms are essential for recognizing $S \simeq Q$ by $d(I(S), I(Q)) = 0$. If the third (triangle) axiom in 1.2(c) fails with any positive error, clustering may not be trustworthy Rass et al. (2022).

Condition 1.2(e) asks for *reconstructable* invariants that can be inverted back to original objects and hence are more practical than a DNA code, which is used for identifying humans in practice (if we forget about identical twins) but a DNA code alone is insufficient yet to grow a living organism.

Problem 1.2 formalizes all verifiable conditions 1.2(a-d,f) for any *discriminative* problem (materials identification) and the first goals 1.2(e-f) of the *generative* problem (designing new materials).

**The contributions** to notoriously hard Problem 1.2 are (1) new higher-order invariants, which distinguished all known counter-examples to the completeness of past descriptors, and (2) the ultra-fast detection of (near-)duplicates in the world's largest databases of experimental materials. The previously unrecognized (near-)duplicates skewed real data but can now be filtered out by continuous invariants for upholding scientific integrity and improving machine learning of materials properties.

## 2 REVIEW OF UNRESOLVED CHALLENGES IN CRYSTAL REPRESENTATIONS

Problem 1.2 makes sense for any objects (finite clouds, graphs) under other practical equivalences (rigid motion excluding reflections) instead of crystals and isometry, respectively. The graph isomorphism problem Grohe & Schweitzer (2020) considers only conditions 1.2(a,b,e,f) without continuous metrics. Boutin & Kemper (2004) proved that pairwise distances distinguish all generic finite clouds of unordered points in $\mathbb{R}^n$. All singular examples within a subspace of measure 0 among all point clouds were distinguished in Widdowson & Kurlin (2023) but we focus on periodic sets.

In 1930, Pauling noticed the ambiguity of crystal structures obtained by diffraction Pauling & Shappell (1930), which called for stronger invariants. For $n = 1$, Theorem 4 in Grünbaum & Moore (1995) justified complete invariants for periodic sequences given by rational angles of the unit circle (in the complex plane $\mathbb{C}$) by using 6-factor products of complex numbers. Since the circle (a period) was fixed, these invariants are discontinuous under perturbations. Indeed, the sequence $\mathbb{Z}$ of integers can be infinitely close to $S = \{0, 1 + \varepsilon, \ldots, m + \varepsilon\} + (m + 1)\mathbb{Z} \subset \mathbb{R}$ for small $\varepsilon > 0$, though their periods 1 and $m + 1$ are arbitrarily different. The much simpler complete invariant of a periodic sequence $S = \{p_1, \ldots, p_m\} + L\mathbb{Z} \subset \mathbb{R}$ with a period $L$, where $0 \leq p_1 < \cdots < p_m < L$, is the list of interpoint distances $p_{i+1} - p_i$ (up to cyclic permutations) for $i = 1, \ldots, m$ and $p_{m+1} = p_1 + L$.

A continuous metric $d(S, Q)$ on these cyclic classes of distance lists was introduced in Kurlin (2022) but such a metric requires an expansion to the least common multiple of the sizes $|S|, |Q|$ of motifs and doesn't come with a polynomial-time invariant. The brute force invariant for all periodic sequences $S$ with motifs up to $m$ points needs an expansion to at least $2^m$ points, see Theorem 5(1)in Farhi (2007), which violates condition 1.2(e). So Problem 1.2 remained open even for $n = 1$.

A finite approach to measuring the similarity between periodic point sets is to compare their finite subsets within a box or a ball of a large but fixed cut-off radius. However, any periodic point set has many non-isometric finite subsets within differently positioned boxes or balls, see Fig. 1 (left).

Considering local clouds centered at all points in a motif $M$ gives invariants such as MACE Batatia et al. (2022), which achieved excellent results by training on large datasets. Perturbing a cut-off radius can discontinuously change these clouds by including new neighbors that were just outside a smaller cut-off. Even if this cut-off is smoothed out, any fixed size is insufficient Parsaeifard & Goedecker (2022), Pozdnyakov et al. (2022): "indistinguishable configurations affect the expressive power of models based on those features, which will be incapable of predicting distinct values for the corresponding atom-centered properties, *even if both structures are used during training*."

Atomic vibrations are natural to measure by deviations of atoms from their initial positions but a sum of small deviations over infinitely many points can be infinite and also can give different values for different finite subsets. However, a maximum deviation of atoms is well-defined as the bottleneck distance between any sets via bijections between atoms, which can be displaced but cannot vanish.

**Definition 2.1.** *The* bottleneck distance $d_B(S, Q) = \inf\limits_{g:S \to Q} \sup\limits_{p \in S} |p - g(p)|$ *for any sets* $S, Q \subset \mathbb{R}^n$

*of the same cardinality is minimized for all bijections* $g : S \to Q$ *and maximized for all* $p \in S$. ∎

Here $|p - q|$ denotes Euclidean distance between points $p, q \in \mathbb{R}^n$. The bottleneck distance $d_B(S, Q)$ is infinite if periodic point sets $S, Q$ have different point densities (motif size $|S|$ divided by the cell volume). Also, $d_B(S, Q)$ is discontinuous under perturbations of 2D lattices whose *primitive* cells have the same minimum volume, see Examples 2.1 and 2.2 in Widdowson & Kurlin (2022). Hence condition 1.2(d) of a Lipschitz continuous metric made Problem 1.2 exceptionally hard.

**Definition 2.2** (*metrics* and *pseudo-metrics*). *A distance $d$ between objects with an equivalence relation $\sim$ is called a* metric *if these axioms hold: (1) $d(S, Q) = 0$ if and only if $S \sim Q$; (2) $d(S, Q) = d(Q, S)$; (3) $d(S, Q) + d(Q, T) \geq d(S, T)$. If axiom (1) is replaced with $(1')$ $d(S, S) = 0$ for any $S$, then non-equivalent $S \not\sim Q$ can have $d(S, Q) = 0$, and $d$ is called a* pseudo-metric. ∎

Many descriptors are compared by distances (such as Euclidean) that satisfy metric axioms on invariant values but define only pseudo-metrics on isometry classes because of incompleteness of the underlying invariants. If $d(S, Q) > 0$, then $S \not\sim Q$ by $(1')$, so a fast pseudo-metric can distinguish between some but not all objects. Pseudo-metrics are weaker than metrics, e.g. the difference $||S| - |Q||$ of the set sizes is a pseudo-metric not distinguishing any sets $S \neq Q$ of the same size.

Metrics (similar to complete invariants) are much more valuable than pseudo-metrics (similar to non-invariants or incomplete invariants). Any algorithm using an incomplete invariant $I$ cannot predict different properties of a *false positive* pair of non-isometric sets $S \not\cong Q$ with $I(S) = I(Q)$.

Hence the *discriminative* problem should be solved first by (at least generically) complete and Lipschitz continuous invariants before any *generative* attempts can succeed. Any non-complete invariant $I$ is not invertible so that different $S \not\cong Q$ can be randomly chosen if $I(S) = I(Q)$. We recall the recent invariants that satisfied almost all conditions 1.2(a-f) for finite and periodic point sets.

**Definition 2.3** (Pointwise Distance Distribution PDD). *Let $S \subset \mathbb{R}^n$ be a periodic point set with a motif $M$ of $m$ points. For any integer $k \geq 1$ and $p \in M$, let $d_1(p) \leq \cdots \leq d_k(p)$ be the list of Euclidean distances from $p$ to its $k$ nearest neighbors within the whole set $S$. These lists become rows of the $m \times k$ matrix $D(S; k)$. Any $l > 1$ identical rows are collapsed into a single row with the weight $l/m$, which is written in the extra first column. The resulting matrix $\mathrm{PDD}(S; k)$ of unordered rows with weights is the* Pointwise Distance Distribution*, see Widdowson & Kurlin (2022).* ∎

If a unit cell of $S$ is extended by a factor of $l$, then any point $p$ in the original motif has $l$ translationally equivalent copies in the extended motif. Then $D(S; k)$ has $l$ times more rows only because each original row is expanded into $l$ identical rows. The final $\mathrm{PDD}(S; k)$ is the same weighted distribution of rows, independent of an initial cell of $S$. The equality between weighted distributions is interpreted as a bijection between unordered sets respecting all weights. This equality is best checked not by considering all bijections but by a metric that vanishes only on equal distributions by the first metric axiom. The PDD is Lipschitz continuous, computable in near-linear time (for a fixed dimension) in both $k$ and motif size $m$, and distinguishes all non-isometric sets in *general position* (away form a measure 0 subspace), see Theorems 3.2, 4.3, 4.4, 5.1 in Widdowson & Kurlin (2022).

**Definition 2.4** (homometric sets). *Finite or periodic sets $S, Q \subset \mathbb{R}^n$ are called* homometric *Patterson (1939) if they have the same* Pair Distribution Function (PDF)*, which is a sequence of all inter-point distances of $S$, equivalent to a powder diffraction pattern without a cut-off radius.* ∎

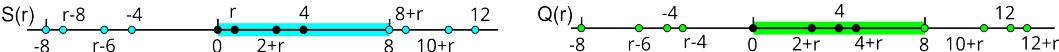

Figure 2: For any $0 < r \leq 1$, the homometric sets $S(r) = \{0, r, 2+r, 4\} + 8\mathbb{Z} \not\cong Q(r) = \{0, r, 2+r, 4\} + 8\mathbb{Z}$ have identical PDFs from Definition 2.4 but different PDDs whose first columns we write as unordered sets: $\mathrm{PDD}(S(r); 1) = \{r, r, 2-r, 2-r\} \neq \mathrm{PDD}(Q(r); 1) = \{r, r, 2-r, 2+r\}$.

**Example 2.5** (sets with equal PDDs). *The sets $S \not\cong Q$ in (Pozdnyakov & Ceriotti, 2022, Fig. 4) were designed to fail all iterations of the Weisfeiler-Leman test Shervashidze et al. (2011). Fig. 3 shows their 2D versions with period 4 in the $x$-axis and free parameters $a, b, c > 0$.*

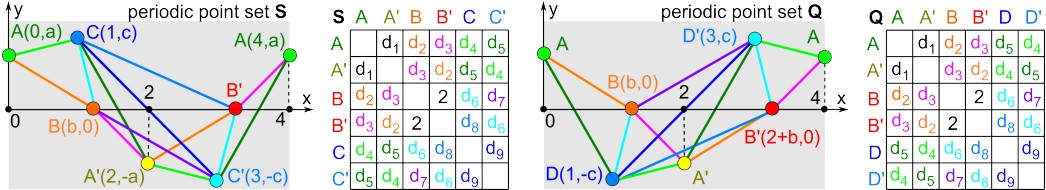

Figure 3: The sets $S, Q$ are 1-periodic in the $x$-axis with period 4, e.g. $A$ denotes both $(0, a)$, $(4, a)$. **Right**: the matrices of distances between closest points from classes modulo shifts by 4 in $x$. Then $\mathrm{PDD}(S; k) = \mathrm{PDD}(Q; k)$ by Example 2.5 but $\mathrm{PDD}^{\{2\}}(S; 1) \neq \mathrm{PDD}^{\{2\}}(Q; 1)$ by Example 3.3.

*The distances in Fig. 3 (right) are for the closest representatives of 6 points, so $d_2 = \sqrt{a^2 + b^2}$,*
$d_1 = 2\sqrt{a^2 + 1}$, $\quad d_3 = \sqrt{a^2 + (2-b)^2}$, $\quad d_4 = \sqrt{1 + (a-c)^2}$, $\quad d_6 = \sqrt{(1-b)^2 + c^2}$,
$d_9 = 2\sqrt{c^2 + 1}$, $\quad d_7 = \sqrt{(3-b)^2 + c^2}$, $\quad d_5 = \sqrt{1 + (a+c)^2}$, $\quad d_8 = \sqrt{(1+b)^2 + c^2}$.

*Then $\mathrm{PDD}(S; k) = \mathrm{PDD}(Q; k)$ because the coincidences of distances in Fig. 3 (right) hold after adding any periodic translation, so if $d_1 = d_2$ then $\sqrt{d_1^2 + (4n)^2} = \sqrt{d_2^2 + (4n)^2}$ for $n \in \mathbb{Z}$.* ∎

Pauling & Shappell (1930) described a pair of real homometric crystals, each having 24 atoms in a cubic cell, with equal PDFs and (as it turned out recently) equal PDDs. The simpler non-isometric

finite sets in $\mathbb{R}^3$ with equal PDDs were distinguished by stronger invariants in Widdowson & Kurlin (2023), which extended PDD by recording distances to subsets of more than one point. In the periodic case, pairs of points behave discontinuously under cell extensions in Fig. 1. Doubling a motif $M$ of $m$ points leads to $(2m)^2$ pairs including new distant neighbors from adjacent cells. This crucial obstacle motivated a 'pointwise' approach to both finite and periodic sets in the next section.

## 3 THE NEW STRONGER $\mathrm{E}(n)$-INVARIANTS OF FINITE AND PERIODIC SETS

Infinitely many pairs of non-isometric sets $S \not\simeq Q$ with equal $\mathrm{PDD}(S;k) = \mathrm{PDD}(Q;k)$ in Example 2.5 motivated the new stronger invariants $\mathrm{PDD}^{\{h\}}$ below. Definition 3.1 makes sense for a finite set $S = M$ in any metric space. The invariant PDD in Definition 2.3 is the case of order $h = 1$.

**Definition 3.1** (higher-order $\mathrm{PDD}(S;k_1,\ldots,k_h)$). *Let $S$ be a periodic point set with a motif $M$ of $m$ points in $\mathbb{R}^n$. Fix a point $p \in M$, integers $h \geq 1$ and $k_1,\ldots,k_h \geq 1$. Consider any $h$ distinct points $p_1,\ldots,p_h \in S - \{p\}$ and the $h$-order average $\dfrac{2}{h(h+1)} \sum_{0 \leq i < j \leq h} |p_i - p_j|$ of pairwise distances between the points $p = p_0, p_1,\ldots,p_h \in S$. Extend the row of $p$ in the $m \times k_1$ matrix $D(S;k_1)$ from Definition 2.3 by writing the $k_2$ smallest 2-order averages $a(p;1) \leq \cdots \leq a(p;k_2)$, then the $k_3$ smallest 3-order averages and so on up to order $h$. In the resulting $m \times (\sum_{i=1}^{h} k_i)$-matrix, collapse any $l > 1$ equal rows to one row with the* weight $l/m$ *written in the extra first column. The final matrix of rows with weights is the $h$-order Pointwise Distance Distribution $\mathrm{PDD}(S;k_1,\ldots,k_h)$. If $(k_1,\ldots,k_h) = (0,\ldots,0,k)$, the brief notation is $\mathrm{PDD}^{\{h\}}(S;k)$. If $k_1 = \cdots = k_h = k$, the $m \times (kh)$-matrix $\mathrm{PDD}^{(h)}(S;k) := \mathrm{PDD}(S;k,\ldots,k)$ consists of the sequentially written $k \times h$ matrices $\mathrm{PDD}^{\{1\}},\ldots,\mathrm{PDD}^{\{h\}}$. Then $\mathrm{PDD}^{\{1\}} = \mathrm{PDD}^{(1)}$ is PDD from Definition 2.3.* ∎

**Example 3.2** ($\mathrm{PDD}^{\{2\}}$ for the sequences in Fig. 2). *The sum $\sum_{0 \leq i < j \leq 2} |p_i - p_j|$ is the perimeter of the triangle on the points $p_0 \in M$ and $p_1, p_2 \in S$. The row of a point $p \in M$ in $\mathrm{PDD}^{\{2\}}(S;k)$ consists of the $k$ smallest perimeters (divided by 3) of triangles at the common vertex $p$. In Fig. 2, the point $p_0 = 0$ in the motif of $S(r) = \{0, r, 2 + r, 4\} + 8\mathbb{Z}$ has nearest neighbors $p_1 = r$, $p_2 = 2 + r$ at the distances $r, 2 + r$, and two smallest averaged perimeters $2(2+r)/3, 8/3$. The point $p_0 = 0$ in $Q(r) = \{0, r, 2+r, 4\} + 8\mathbb{Z}$ has nearest neighbors at the distances $2 + r, 4 - r$, and two smallest averaged perimeters $\frac{8}{3}, \frac{8}{3}$. The computations for other points give $\mathrm{PDD}(S(r);2,2) =$*

$$
\begin{pmatrix}
r & 2+r & \frac{2(2+r)}{3} & \frac{8}{3} \\
r & 2 & \frac{2(2+r)}{3} & \frac{2(4-r)}{3} \\
2-r & 2 & \frac{2(2+r)}{3} & \frac{2(4-r)}{3} \\
2-r & 4-r & \frac{2(4-r)}{3} & \frac{8}{3}
\end{pmatrix}
\neq \mathrm{PDD}(Q(r);2,2) =
\begin{pmatrix}
2+r & 4-r & \frac{8}{3} & \frac{8}{3} \\
2-r & 2+r & \frac{4}{3} & \frac{8}{3} \\
r & 2-r & \frac{4}{3} & \frac{8}{3} \\
r & 2 & \frac{4}{3} & \frac{8}{3}
\end{pmatrix}.
$$
∎

The factor $\dfrac{2}{h(h+1)}$ was chosen to guarantee the Lipschitz continuity with $\lambda = 2$ in (1.2d). Our experiments use $h = 2, 3$ to substantially strengthen PDD of order $h = 1$. Indeed, Examples 3.3, 3.6 will show that $\mathrm{PDD}^{(2)}$ distinguishes all known homometric sets for $n = 2, 3$. The numbers $k_1,\ldots,k_h$ are usually chosen equal: $k = k_1 = \cdots = k_h$. Any increase in this number $k$ of nearest neighbors only adds larger values to the $\mathrm{PDD}^{\{h\}}$ invariants without changing any of the previous values. Hence $k$ is considered a degree of approximation, not a parameter like a cut-off radius whose changes can substantially affect local atomic environments. If an atom has different neighbors at equal distances (or nearly equal up to $\varepsilon$), the order (hence positions) of these neighbors can be discontinuously swapped under perturbation but the distances change continuously up to $2\varepsilon$.

**Example 3.3** ($\mathrm{PDD}^{\{2\}}$ distinguishes $S, Q$ in Example 2.5). *We start with singular cases. If $c = 0$, then $C = D$, $C' = D'$, so $S, Q$ are identical in Fig. 3. If $b \in \{0, 1, 2\}$, periodic shifts of $B \cup B'$ (hence $S, Q$) become mirror images with respect to the vertical line $x = 2$. In all other cases, Example B.1 in the appendix checks that the smallest perimeter of triangles on points of $S$ differs from the smallest perimeter for $Q$. Then $\mathrm{PDD}^{\{2\}}(S;1) \neq \mathrm{PDD}^{\{2\}}(Q;1)$ and hence $S \not\simeq Q$.* ∎

Because $\text{PDD}^{\{h\}}$ has ordered columns (by the index $k$ of neighbors) and unordered rows (representing points in a motif), all such matrices even with different numbers of rows can be compared by Earth Mover's Distance, see Definition 3.5. We can convert any $\text{PDD}^{\{h\}}$ into a fixed-size matrix, which can be flattened into a vector for easy comparisons, while keeping the continuity and almost all invariant data. Any distribution of $m$ unordered values can be reconstructed from its $m$ moments defined below. When all weights $w_i$ are rational as in our case, the distribution can be expanded to equal-weighted values $a_1, \ldots, a_m$. The $m$ moments can recover all $a_1, \ldots, a_m$ as roots of a polynomial of degree $m$ whose coefficients are expressed via the $m$ moments Macdonald (1998). For example, any reals $a, b$ are the roots of $t^2 - (a+b)t + ab$, where $ab = \frac{1}{2}((a+b)^2 - (a^2 + b^2))$.

Let $A$ be any unordered set of real numbers $a_1, \ldots, a_m$ with weights $w_1, \ldots, w_m$, respectively, such that $\sum_{i=1}^{m} w_i = 1$. For any integer $l \geq 1$, the $l$-th *moment* (Keeping, 1995, section 2.7) is

$$\mu_l(A) = \sqrt[l]{m^{1-l} \sum_{i=1}^{m} w_i a_i^l}, \text{ so } \mu_1(A) = \sum_{i=1}^{m} w_i a_i \text{ is the usual average. For } l \geq 2, \text{ we normalize by}$$

the factor $m^{(1/l)-1}$ to prove the continuity of all moments with the Lipschitz constant $\lambda = 2$.

**Definition 3.4** (Pointwise Distance Moment PDM[$l$]). *Fix integers $l, h \geq 1$. For a column $A$ of the Pointwise Distance Distribution $\text{PDD}(S; k_1, \ldots, k_h)$, which consists of unordered numbers $a_1, \ldots, a_m$ with weights from Definition 3.1, write the new column $(\mu_1(A), \ldots, \mu_l(A))$. The new $l \times (\sum_{i=1}^{h} k_i)$ matrix is the Pointwise Distance Moment $\text{PDM}[l](S; k_1, \ldots, k_h)$. Then $\text{PDM}[1](S; k)$ is called the vector of Average Minimum Distances $\text{AMD}(S; k) = (\text{AMD}_1, \ldots, \text{AMD}_k)$.* ∎

The matrix PDM[$l$] has ordered rows and columns but is a bit weaker than PDD (with the same $h, k_1, \ldots, k_h$) because each column is reconstructable from its moments (for large enough $l$) only up to permutation, but PDM[$l$] more quickly filters distant crystals. We can flatten any matrix PDM[$l$] with indexed entries to a vector. Vectors $u, v \in \mathbb{R}^m$ of distances are compared by $L_\infty(u, v) = \max_{i=1,\ldots,m} |u_i - v_i|$ which controllably changes under perturbations of interatomic distances.

**Definition 3.5** (Earth Mover's Distance EMD Rubner et al. (2000)). *Let a $X$ be a space with a base metric $d$. Any unordered set $\{(R_i, w_i)\}_{i=1}^{m}$ of objects $R_i \in X$ with weights $w_i > 0$ such that $\sum_{i=1}^{m} w_i = 1$ is called a (normalized)* weighted *distribution. For any such distributions $A = \{(R_i(A), w_i(A))\}_{i=1}^{m(A)}$ and $B = \{(R_i(B), w_i(B))\}_{i=1}^{m(B)}$, the Earth Mover's Distance $\text{EMD}(A, B) = \min_{f_{ij}} \sum_{i=1}^{m(A)} \sum_{j=1}^{m(B)} f_{ij} d(R_i(A), R_j(B))$ is minimized for all real $f_{ij} \geq 0$ (called* flows*) subject to the conditions $\sum_{i=1}^{m(A)} f_{ij} \leq w_j(B)$, $\sum_{j=1}^{m(B)} f_{ij} \leq w_i(A)$, $\sum_{i=1}^{m(A)} \sum_{j=1}^{m(B)} f_{ij} = 1$.* ∎

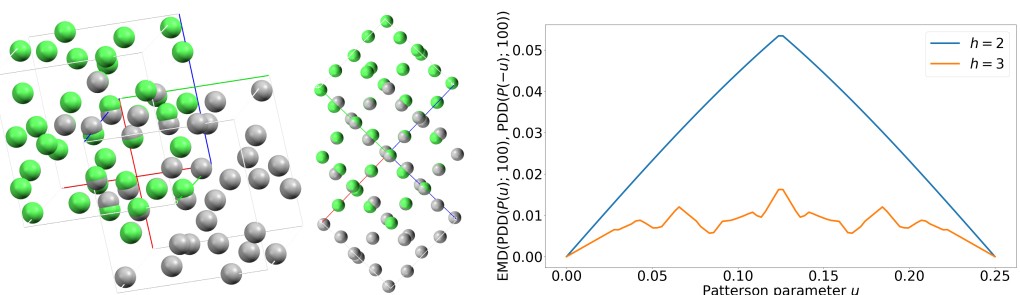

Figure 4: **Left**: a comparison of Pauling's homometric crystals $P(\pm u)$ for $u = 0.03$ Pauling & Shappell (1930), by COMPACK Chisholm & Motherwell (2005), which aligns subsets of 15 (default, left) atoms and 48 (twice the size of the motif, right). The atoms from different $P(\pm 0.03)$ are shown in green and gray. **Right**: EMD is between $\text{PDD}^{\{h\}}$ for $k = 100$ and Pauling's crystals $P(\pm u)$, which continuously depend on $u \in [0, 0.25]$ and are identical at the boundary values.

**Example 3.6** (ablation study). *Fig. 4 (left) shows a pair of overlaid Pauling crystals $P(\pm 0.03)$ with 24 atoms in a cubic cell Pauling & Shappell (1930). The importance of $\mathrm{PDD}^{\{2\}}$ in comparison with the past invariants $\mathrm{PDD}$ is demonstrated by the infinite series of periodic sets $P(\pm u) \subset \mathbb{R}^3$, which have the same $\mathrm{PDD}(P(u); k) = \mathrm{PDD}(P(-u); k)$ for all parameters $u \in (0, 0.25)$ and $k \geq 1$ but different $\mathrm{PDD}^{\{2\}}(S; 100)$ and $\mathrm{PDD}^{\{3\}}(S; 100)$ due to distances $\mathrm{EMD} > 0$ in Fig. 4 (right).* ∎

## 4 LIPSCHITZ CONTINUOUS METRIC, ASYMPTOTIC, AND TIME OF $\mathrm{PDD}^{\{h\}}$

This section states the key properties of $\mathrm{PDD}^{\{h\}}$: the important Lipschitz continuity in Theorem 4.1, Theorem 4.3 solving Problem 1.2 for $n = 1$, asymptotic Theorem 4.4, and hardest Theorem 4.5.

For any discrete set $S$, the *packing radius* $r(S)$ is the minimum half-distance between points of $S$.

**Theorem 4.1** (Lipschitz continuity). *Fix integers $h, k_1, \ldots, k_h, l \geq 1$. If each point of a finite or periodic point set $S$ is perturbed up to a distance $\varepsilon \in [0, r(S))$, both $\mathrm{PDD}(S; k_1, \ldots, k_h)$ and $\mathrm{PDM}[l](S; k_1, \ldots, k_h)$ change by at most $2\varepsilon$ in the metrics $\mathrm{EMD}$ and $L_\infty$, respectively.* ∎

Fig. 5 shows how $\mathrm{EMD}$ between $\mathrm{PDD}^{\{2\}}$s continuously changes under perturbations of sets.

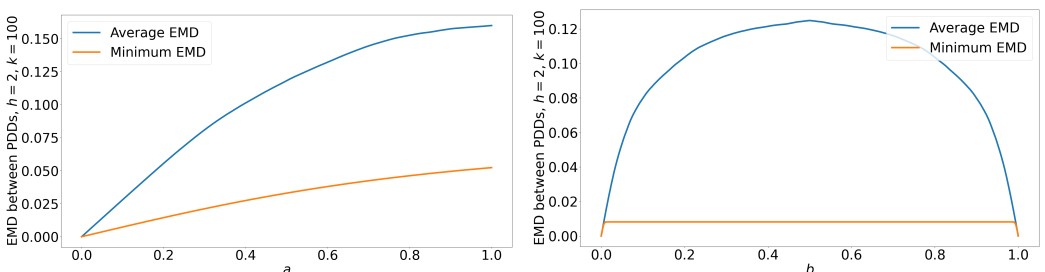

Figure 5: Distance metric $\mathrm{EMD}$ between $\mathrm{PDD}^{\{2\}}$ for $k = 100$ and the homometric 1-periodic sets $S, Q$ with uniformly sampled $a, b, c$ in Fig. 3. These sets $S, Q$ are isometric for $b \in \{0, 1\}$ but $\mathrm{EMD} > 0$ for all $0 < b < 1$, which experimentally confirms the proof that $S \not\simeq Q$ in Example 3.3.

For a finite set $S \subset \mathbb{R}$, a simple complete invariant under translations is the ordered sequence of inter-point distances but its naive extension to periodic sets is discontinuous. Any hopeful attempt at Problem 1.2 should start from dimension $n = 1$, which is finally solved by Theorem 4.3 below.

**Definition 4.2** (*Pointwise Shift Distribution* PSD). *For any periodic point set (sequence) $S \subset \mathbb{R}$ with a motif $M$ of $m$ points, write down distances from each $p \in M$ to its $k$ nearest neighbors $q > p$ in increasing order in a row of an $m \times k$-matrix. Collapse any $l > 1$ equal rows to one row with the weight $l/m$ in an extra first column. The resulting matrix $\mathrm{PSD}(S; k)$ is called the Pointwise Shift Distribution, which also makes sense for any finite set $S = M \subset \mathbb{R}$ of unordered points.* ∎

**Theorem 4.3.** *For all finite sets $M \subset \mathbb{R}$ of $m$ unordered points, $\mathrm{PSD}(M; m-1)$ solves Problem 1.2. For all periodic point sets $S \subset \mathbb{R}$ with $m$ points in a motif, $\mathrm{PSD}(S; m)$ solves Problem 1.2.* ∎

To analyze $\mathrm{PDD}^{\{h\}}(S; k)$ as $k \to +\infty$, for $h, k \geq 1$, choose a real $b \geq h$ such that $\binom{b}{h} = \dfrac{b(b-1) \ldots (b-h+1)}{h!}$ belongs to $(k-1, k]$. Set $b(h, k) = b + 1$ e.g. $b(1, k) = k + 1$, $b(2, k) = 1.5 + \sqrt{2k}$. Let $V_n$ be the unit ball volume in $\mathbb{R}^n$. Any periodic set $S \subset \mathbb{R}^n$ with a motif of $m$ points and unit cell of volume $\mathrm{vol}[U]$ has the *point packing coefficient* $\mathrm{PPC}(S) = \sqrt[n]{\dfrac{\mathrm{vol}[U]}{mV_n}}$.

**Theorem 4.4** (asymptotic of $\mathrm{PDD}^{\{h\}}$). *Let a periodic point set $S \subset \mathbb{R}^n$ have a cell with a longest diagonal $d$. For $h, k \geq 1$, let $a(h, k)$ be an average sum in the $k$-th column of $\mathrm{PDD}^{\{h\}}(S; k)$. Then $\dfrac{2}{h+1}\left(\mathrm{PPC}(S)\sqrt[n]{b(h,k)} - d\right) \leq a(h, k) \leq \dfrac{2h}{h+1}\left(\mathrm{PPC}(S)\sqrt[n]{b(h,k)} + d\right)$ for $k \geq 1$. If $h = 1$, $\lim\limits_{k \to +\infty} \dfrac{a(1, k)}{\sqrt[n]{k}} = \mathrm{PPC}(S)$. If $h = 2$, $\dfrac{2}{3}\mathrm{PPC}(S) \leq \dfrac{a(2, k)}{\sqrt[2n]{2k}} \leq \dfrac{4}{3}\mathrm{PPC}(S)$ for all big enough $k$.* ∎

Theorem 4.4 illustrated in Fig. 15 justifies that there is no need to substantially increase the number $k$ of neighbors since $\mathrm{PDD}^{\{h\}}(S;k)$ largely depends on the point packing coefficient $c(S)$ when $k \to +\infty$. The practical advice is to choose $k$ depending on the size of a motif or constituent molecule so that all atoms have enough neighbors to capture the periodic connectivity. We consider $k$ a degree of approximation similar to the number of decimal places on a calculator. Theorem 4.4 implies similar bounds for all moments from $\mathrm{PDM}^{\{h\}}[l]$ and means that $\mathrm{PDD}^{\{h\}}(S;k)$ and $\mathrm{PDM}^{\{h\}}[l](S;k)$ are most discriminative for small values of $k$, so we used $k = 100$ and $l = 10$ in our experiments.

**Theorem 4.5** (time of $\mathrm{PDD}^{\{h\}}$). *For any $h, k \geq 1$ and a periodic point set $S \subset \mathbb{R}^n$ with a motif of $m$ points and a unit cell $U$ with a longest diagonal $d$ and skewness $\nu(U) = \dfrac{d}{\sqrt[n]{\mathrm{vol}[U]}}$, the number of arithmetic operations to compute $\mathrm{PDD}^{\{h\}}(S;k)$ is proportional to at most $mN \log N$ with $N \leq$*

$$\frac{2^h}{h!}(2h+3)^{hn}\big((2h+3)^h k + (V_n \nu(U)m)^{hn}\big), \text{ linear in } k, \text{ polynomial in } m \text{ for fixed } n, h, \nu(U).$$

## 5 EXPERIMENTS ON THE WORLD'S FIVE LARGEST DATABASES OF CRYSTALS

This section adapts the new invariants to average summaries in Definition 5.1 and report thousands of previously unknown (near-)duplicates in the five world's largest public databases Taylor & Wood (2019); Gražulis et al. (2009); Zagorac et al. (2019); Jain et al. (2013); Merchant et al. (2023). The sizes in Table 1 below are the numbers of all periodic crystals (no disorder and full geometric data) in September 2024 (total number 1,818,588), see more details of all experiments in appendix A.

Table 1: Links and sizes (numbers of pure periodic crystals) of the world's five largest databases.

| database | crystals | web address |
|---|---|---|
| CSD : Cambridge Structural Database | 831,126 | ccdc.cam.ac.uk/solutions/software/csd |
| COD : Crystallography Open Database | 344,127 | www.crystallography.net/cod |
| ICSD : Inorganic Crystal Struct. Database | 105,162 | icsd.products.fiz-karlsruhe.de/en |
| MP : Materials Project by the Berkeley lab | 153,235 | next-gen.materialsproject.org |
| GNoME : Graph Net. Materials Exploration | 384,938 | github.com/google-deepmind/materials_discovery |

To neutralize the effect of increasing distances $\mathrm{AMD}_k$ with respect to $k$, Theorem 4.4 motivated subtract the asymptotic $c(S)\sqrt[3]{k}$ in Definition 5.1 for the invariants ADA. Fig. 6 shows how the purely geometric information easily differentiated between organic-vs-inorganic databases. For all crystals, $\mathrm{ADA}_k$ decrease to 0 as $k \to +\infty$ justifying our computations up to $k = 100$ below.

**Definition 5.1** (Average/Pointwise Deviations from Asymptotic: ADA, PDA). *Distances in $\mathrm{PDD}(S;k)$ are increasing in $k$ by Theorem 4.4, to avoid the dominance by the largest value of $k$, the vector $\mathrm{ADA}(S;k)$ and matrix $\mathrm{PDA}(S;k)$ are obtained from $\mathrm{AMD}(S;k), \mathrm{PDD}(S;k)$ by subtracting $\mathrm{PPC}(S)\sqrt[n]{i}$ from each $i$-th coordinate/column, respectively, for all $i = 1, \ldots, k$.* ∎

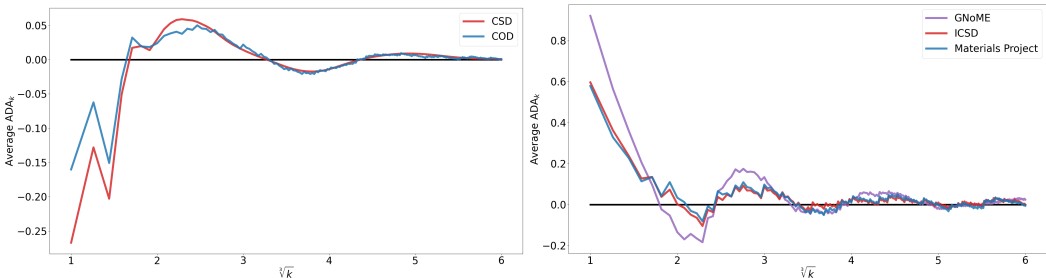

Figure 6: The averages of $\mathrm{ADA}(S;k)$ across a database vs $\sqrt[3]{k}$ easily differentiate between major chemical types. **Left**: mostly organic crystals (of main elements H,C,O,N,S,P) whose lack of symmetry makes the $\mathrm{ADA}_k$ average smooth for $k > 10$. **Right**: mostly inorganic crystals (metals) whose high symmetry (as in cubic table salt NaCl) explains the wiggling of the $\mathrm{ADA}_k$ average.

**Hierarchical computation**. The new higher-order invariants $\mathrm{PDD}^{\{h\}}$ form a natural hierarchy starting from the simpler and faster invariants ADA and PDA. We first used the vector $\mathrm{ADA}(S; 100)$ to find nearest neighbors across all databases by kd-trees Gieseke et al. (2014) up to $L_\infty \leq 0.01\text{Å}$. Since the smallest inter-atomic distances are about $1\text{Å} = 10^{-10}$m, atomic displacements up to $0.01\text{Å}$ are considered experimental noise. For the closest pairs found by $\mathrm{ADA}(S; 100)$, the stronger $\mathrm{PDA}(S; 100)$ can have only larger distances $\mathrm{EMD} \geq L_\infty$ by (Cohen & Guibas, 1997, section 3). The CSD, COD, ICSD are expected to contain only experimental structures, while MP and GNoME are obtained by simulations. Table 2 shows that the well-curated 59-year-old CSD has 0.9% near-duplicate crystals, while more than a third of the ICSD consists of near-duplicates that are geometrically almost identical so that all atoms can be matched by an average perturbation up to $0.01\text{Å}$. (Anosova et al., 2024, section 6) described thousands of more embarrassing exact duplicates, where chemical elements were replaced while keeping all coordinates fixed. These replacements are physically impossible without more substantial perturbations of geometry, so several journals are investigating data integrity Chawla (2024), see more examples in Appendix A.

The bold numbers in Table 2 count near-duplicates within each database, which should be filtered out for any analysis or machine learning else the ground truth data becomes skewed, see the percentages for different thresholds in Fig. 2 (right). Other numbers are matches across different databases.

Table 2: Count and percentage of all pure periodic crystals in each database (left) found to have a near-duplicate in other databases (top) by the distance $\mathrm{EMD} < 0.01\text{Å}$ on matrices $\mathrm{PDA}(S; 100)$.

| databases | CSD | | COD | | ICSD | | MP | | GNoME | |
|---|---|---|---|---|---|---|---|---|---|---|
| | count | % | count | % | count | % | count | % | count | % |
| CSD | **7687** | **0.9** | 272649 | 32.8 | 4649 | 0.6 | 21 | 0.0 | 1 | 0.0 |
| COD | 276328 | 80.3 | **19231** | **5.6** | 36553 | 10.6 | 5239 | 1.52 | 2705 | 0.8 |
| ICSD | 4736 | 4.5 | 48899 | 46.5 | **35189** | **33.5** | 16386 | 15.6 | 9123 | 8.7 |
| MP | 64 | 0.0 | 11989 | 7.82 | 14312 | 9.3 | **19177** | **12.5** | 10681 | 7.0 |
| GNoME | 2 | 0.0 | 1801 | 0.5 | 2459 | 0.6 | 3401 | 0.9 | **82859** | **21.5** |

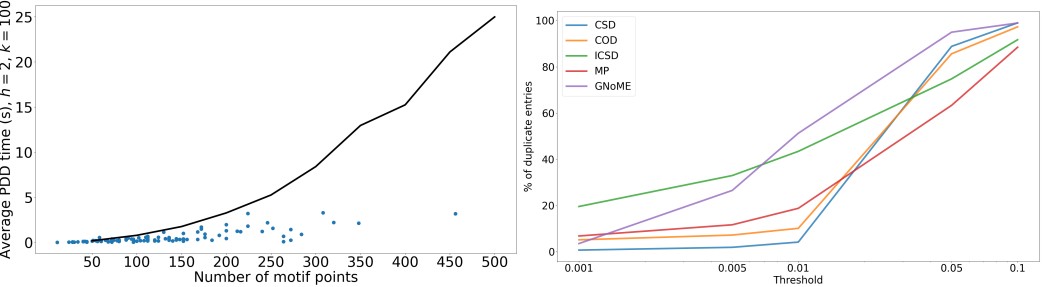

Figure 7: **Left**: Times in seconds for $\mathrm{PDD}^{\{2\}}(S; 100)$ vs the motif size $m$. **Black**: random periodic sets with cell sizes in the range $[1, 2]$ and angles in $[60°, 120°]$. **Blue**: times for real crystals in the CSD. **Right**: growing percentages of near-duplicates in 5 databases for different thresholds in Å.

In the past, the (near-)duplicates were impossible to detect at scale, because the traditional comparison through iterative alignment of 15 (by default) molecules by the COMPACK algorithm Chisholm & Motherwell (2005) is too slow for all-vs-all comparisons. Tables 3 and 4 compare the running times: **hours** of $\mathrm{PDA}(S; 100)$ vs **years** of RMSD, extrapolated for the same machine from the median time 117 ms (average 582 ms) on 500 random pairs in the CSD. On the same 500 pairs, $\mathrm{PDA}(S; 100)$ for two crystals per pair and distance EMD took only 7.48 milliseconds on average. All experiments were done on a typical desktop (AMD Ryzen 5 5600X 6-core, 32GB RAM).

## 6 DISCUSSION OF LIMITATIONS, INTEGRITY, AND GROWING SIGNIFICANCE

For more than 100 years, crystallography relied on determining 3D structures from diffraction patterns. Recently, Shen et al. (2022) showed how to convert any crystal into many different homometric structures indistinguishable from the original by diffraction. Earlier Fig. 1 (right) showed that any known crystal can also be disguised by changing a unit cell, shifting atoms a bit, changing chemical elements, then claimed as 'new', see adversarial Algorithm A.1 in appendix A.

Table 3: Times in seconds (less than 8.5 hours in total) to find near-duplicates in Table 2 with EMD $\leq 0.01$Å on PDA$(S; 100)$ across five major databases, compare with years in Table 4.

| databases | CSD | COD | ICSD | MP | GNoME | sum of times, hrs:min:sec |
|---|---|---|---|---|---|---|
| CSD | 403.6 | 1979.3 | 42.9 | 6.2 | 4.5 | 0:40:36 |
| COD | 1979.3 | 609.7 | 2249.8 | 1525.4 | 234.5 | 1:49:59 |
| ICSD | 42.9 | 2249.8 | 3362.1 | 4428.1 | 819.3 | 2:49:38 |
| MP | 6.2 | 1525.4 | 4428.1 | 4431.8 | 999.9 | 3:09:51 |
| GNoME | 4.5 | 234.5 | 819.3 | 999.9 | 9436.7 | 3:11:35 |

Table 4: These times for all comparisons by COMPACK Chisholm & Motherwell (2005) are extrapolated from the median time of 117 ms on 500 random pairs from the CSD on the same typical desktop, which completed Table 2 of near-duplicates across all five databases within 8.5 hours.

| database | periodic crystals | all unordered pairs | time, milliseconds | hours | years |
|---|---|---|---|---|---|
| CSD | 831,126 | 345,384,798,375 | $4.04 \times 10^{13}$ | 11,225,006 | 1280.5 |
| COD | 344,127 | 59,211,524,001 | $6.93 \times 10^{12}$ | 1,924,375 | 219.7 |
| ICSD | 105,162 | 5,529,470,541 | $6.47 \times 10^{11}$ | 179,708 | 20.5 |
| MP | 153,235 | 11,740,405,995 | $2.75 \times 10^{12}$ | 763,126 | 87.1 |
| GNoME | 384,938 | 74,088,439,453 | $8.67 \times 10^{12}$ | 2,407,874 | 274.8 |

Such artificially generated structures threaten the integrity of experimental databases Chawla (2024), which are already skewed by previously undetectable near-duplicates. These challenges motivated the stronger questions "how much different?" and "what is behind a code?", which were formalized in Problem 1.2 aiming for a continuous parametrization of the space of crystals . One limitation is that a random PDD$^{\{h\}}$ may not be realizable by a real crystal because inter-atomic distances cannot be arbitrary. However, these invariants parametrize the 'universe' containing all known crystals as 'shiny stars' and all not yet discovered crystals hidden in empty spots on the same map, see Fig. 8.

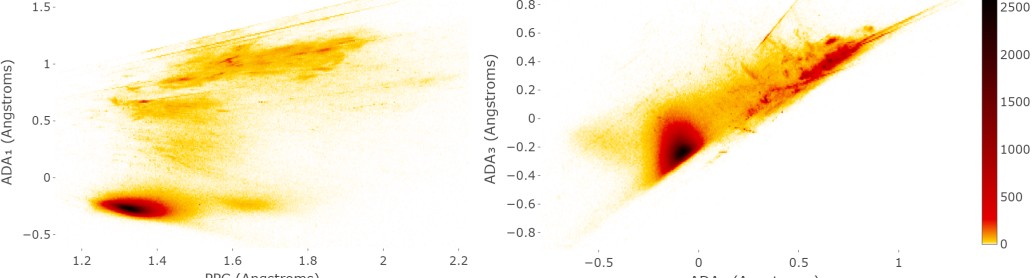

Figure 8: The projections of the five largest databases in the analytically defined invariant coordinates. The color indicates the number of crystals at each location. Experimental crystals occupy the main hot spot, while simulated crystals appear in sharp lines, see more maps in Appendix A.

The new invariants PDD$^{\{h\}}$ complete the hierarchy of the simpler and faster invariants AMD and PDD. While diffraction patterns and PDDs cannot distinguish infinitely many homometric crystals, PDD$^{\{2\}}$ distinguished all known (infinitely many) counter-examples. We use PDD$^{\{2\}}$ only in rare cases to confirm exact duplicates after much faster filtering by ADA, PDA whose times are near-linear in $k, m$ by Theorem 4.5 substantially extending (Widdowson & Kurlin, 2022, Theorem 5.1).

By (Widdowson & Kurlin, 2022, Theorem 4.4), PDD and hence the stronger invariant PDD$^{\{h\}}$ distinguish all crystals in general position. The full completeness of continuous invariants was open even in dimension $n = 1$ Franses & Paap (2004), now complete by Theorem 4.3. The key impact is the efficient barrier for homometric or noisy disguises of known crystals because the invariants can quickly find all nearest neighbors of any newly claimed material in the existing databases.

We thank all reviewers for supporting scientific integrity, now guaranteed by the proposed invariants.

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

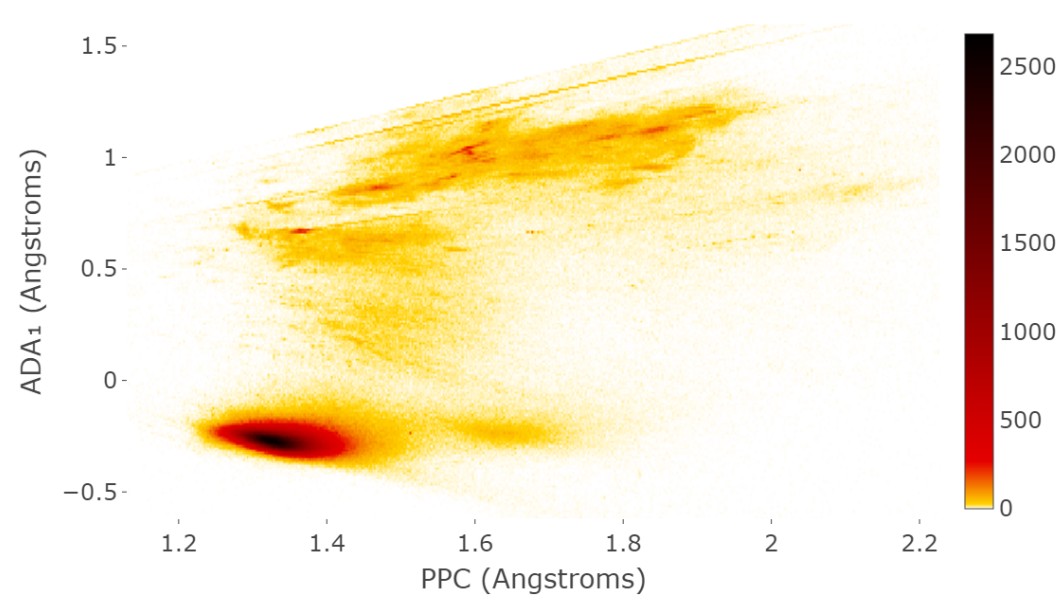

Figure 9: The projection of the world's five largest databases in the invariants PPC (Point Packing Coefficient) and $ADA_1$ (Average Deviation from Asymptotic) from Theorem 4.4 and Definition 5.1.

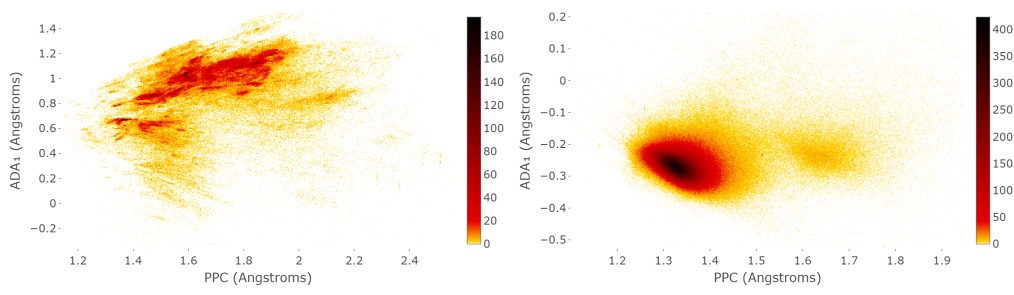

Figure 10: The projections of the GNoME (left) and CSD (right) in the invariants PPC and $ADA_1$.

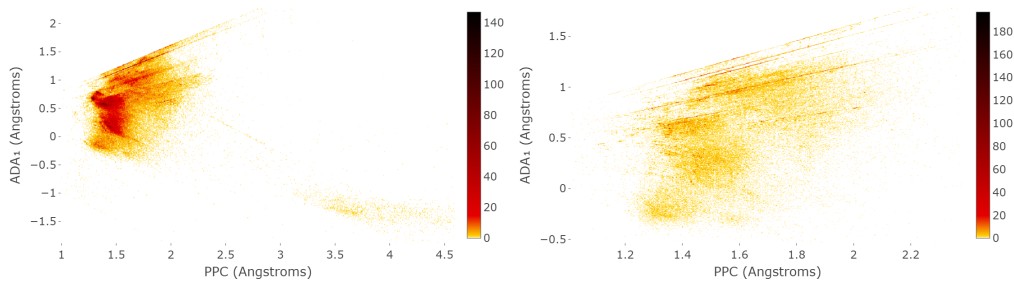

Figure 11: The projections of the MP (left) and ICSD (right) in the invariants PPC and $ADA_1$.

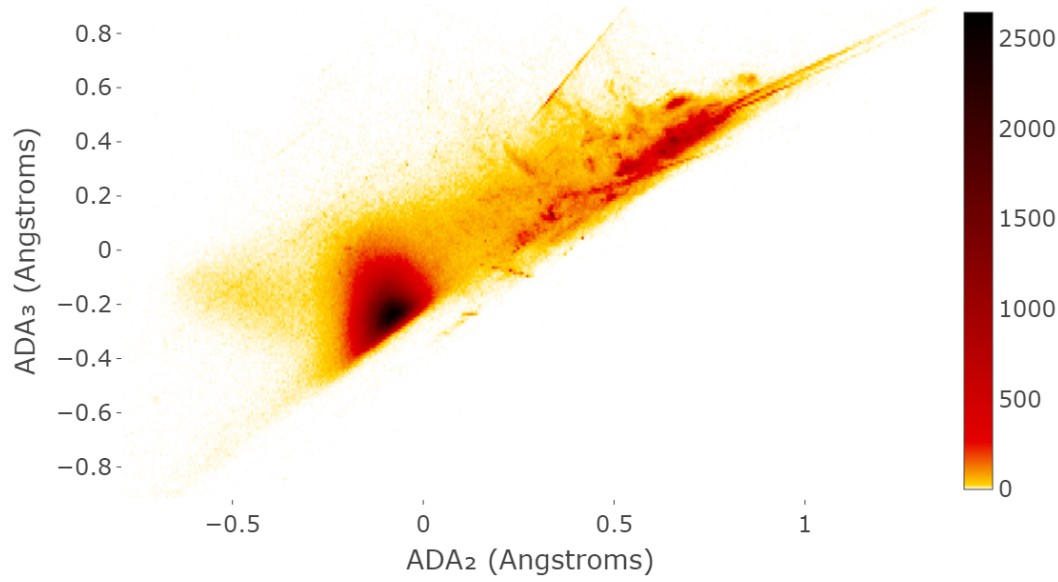

Figure 12: The projection of the world's five largest databases in the invariants $ADA_2$ and $ADA_3$.

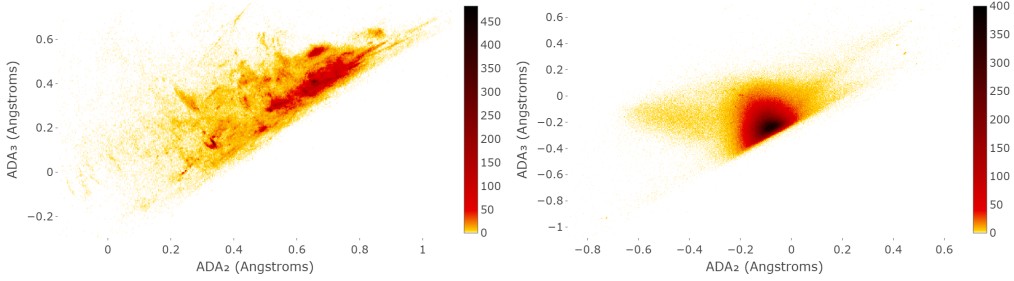

Figure 13: The projections of the GNoME (left) and CSD (right) in the invariants $ADA_2$ and $ADA_3$.

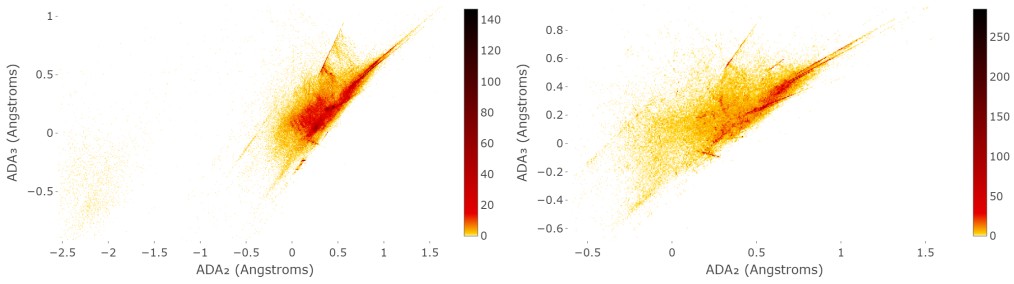

Figure 14: The projections of the MP (left) and ICSD (right) in the invariants $ADA_2$ and $ADA_3$.

## A  APPENDIX: DETAILS OF EXPERIMENTS ON THE FIVE DATABASES

This appendix describes the main experiments in more detail. All sharp lines in Fig. 9 and further maps indicate families of crystals with a specific geometry, for example, cubic crystals whose full geometry and hence all invariants depend on a single parameter (the smallest inter-atomic distance).

Some entries in the CSD and COD are incomplete or disordered (not periodic). After removing such entries, we were left with 831,126 CSD structures and 344,127 COD structures.

Firstly, we computed $\text{PDM}[10](S; 100)$ for all entries, taking 27 min 33 sec for the CSD and 12 mins 15 sec for COD (2 ms per structure on average). To find exact matches between databases by PDM, we make use of the $k$-d tree data structure, designed for fast nearest neighbor lookup. A $k$-d tree can be constructed from any collection of vectors, which can then be queried for a number of nearest neighbors of a new vector, using a binary tree style algorithm with logarithmic search time.

We flattened each $\text{PDM}[10](S; 100)$ matrix to a vector with 1000 dimensions, constructed a $k$-d tree for both CSD and COD, then queried the 10 nearest neighbors for each item in the other. If the most distant neighbor for any entry is closer than the threshold $10^{-13}$Å (within floating point error), we extend the search and find more neighbors until all pairs within the threshold are found. We were left with a total of 270,669 matches; an overlap between the databases of one third of the CSD and almost 80% of COD.

| CSD refcode | COD ID | Notes |
|---|---|---|
| LAVFAP | 2001334 | Mixed types in original CIF |
| ZAYRUM | 2003941 | Mixed types in original CIF |
| FONGAQ01 | 2005101 | Mixed types in original CIF |
| TIPYOG | 2005914 | Mixed types in original CIF |
| HABTAF | 2001740 | Mixed types in original CIF |
| AJIRAM01 | 2100097 | Mixed types in original CIF |
| LABSAI | 2001822 | Mixed types in COD CIF |
| DECTAI | 4065524 | Mixed types in COD CIF |
| WATMIO | 4309447 | Mixed types in COD CIF |
| NAJQUK | 4323901 | Mixed types in COD CIF |
| PIHJUL | 4030494 | Mixed types in COD CIF |
| ELOJOE | 4314231 | CSD remarks replaced atom |
| MARSIH | 4321045 | CSD remarks replaced atom |
| KUTWUU | 7126770 | CSD remarks replaced atom |
| XAVDEF | 4103386 | CSD remarks replaced atom |
| JEMLAP | 4101489 | CSD remarks replaced atom |
| QUCXAP | 7117360 | CSD remarks replaced atom |
| PIBTAW | 1505325 | CSD remarks replaced atom |
| UKAXUB | 7234657 | CSD remarks replaced atom |
| POCLOK | 2220314 | COLYEI is a duplicate |
| COLYEI | 8102533 | POCLOK is a duplicate |
| JEPLIA | 2213484 | HIFCAB is a duplicate |
| LALNET | 8102594 | POPCAA is a duplicate |
| SELHAU | 4027023 | One entry is mistaken |
| PINHUP | 1558382 | One entry is mistaken |
| KABHOL | 4113866 | One entry is mistaken |

Table 5: List of 26 matches between the CSD and COD found to have identical geometry but different chemical compositions.

Of particular interest are the 26 pairs which have different compositions, as the impossibility of complex organic structures sharing the exact same geometry but not composition implies an error or labeling issue. The pairs were confirmed as geometric duplicates by the strongest invariants $\text{PDD}^{\{h\}}$ and found to have different compositions for the reasons in Table 5.

- The original Crystallographic Information File (CIF) has atoms simultaneously labeled as two types or disagreement with what is reported in the published paper (6 pairs),
- Atoms are labeled as two types in the COD CIF (5 pairs),
- Geometric duplicates known to the CSD gave a match with different compositions (4 pairs),
- A remark in the CSD entry explains that atoms were replaced in the curation process because the deposited CIF was incorrect (8 pairs),

- The COD and CSD entries disagree for an unknown reason (3 pairs).

In addition to cross-comparing the CSD and COD, we included the ICSD and Materials Project database (MP) and compared them all pairwise, as well as searching for duplicates within each. Tables 6 and 15 below show how many matches were found, and how many also shared the same composition.

| databases | matches | same composition |
|---|---|---|
| CSD vs COD | 270,669 | 270,583 |
| CSD vs ICSD | 3,913 | 3,913 |
| COD vs ICSD | 35,051 | 31,918 |
| COD vs MP | 2 | 2 |
| ICSD vs MP | 17 | 7 |

Table 6: Number of exact matches (PDM within $10^{-13}$Å) between four databases.

| databases | CSD | | COD | | ICSD | | MP | | GNoME | |
|---|---|---|---|---|---|---|---|---|---|---|
| | count | % | count | % | count | % | count | % | count | % |
| CSD | **36269** | **4.4** | 277354 | 33.4 | 4947 | 0.6 | 103 | 0.0 | 3 | 0.0 |
| COD | 277977 | 80.8 | **30786** | **8.9** | 37233 | 10.8 | 6743 | 2.0 | 3091 | 0.9 |
| ICSD | 5033 | 4.8 | 51604 | 49.1 | **42686** | **40.6** | 20209 | 19.2 | 10605 | 10.1 |
| MP | 152 | 0.1 | 14066 | 9.2 | 17550 | 11.5 | **28806** | **18.8** | 13362 | 8.7 |
| GNoME | 9 | 0.0 | 2768 | 0.7 | 4452 | 1.2 | 11124 | 2.9 | **197340** | **51.3** |

Table 7: Count and percentage of all pure periodic crystals in each database (left) found to have a near-duplicate in other databases (top) by the distance $L_\infty < 0.01$Å on vectors ADA($S$; 100).

| | CSD | COD | ICSD | MP | GNoME | time (s) |
|---|---|---|---|---|---|---|
| CSD | 235.15 | 180.04 | 27.28 | 29.88 | 13.05 | 485.40 |
| COD | 146.92 | 66.33 | 13.38 | 12.79 | 9.57 | 248.99 |
| ICSD | 4.21 | 5.70 | 5.99 | 5.37 | 6.41 | 27.68 |
| MP | 6.30 | 7.48 | 10.19 | 9.32 | 10.17 | 43.46 |
| GNoME | 6.22 | 9.66 | 10.44 | 9.39 | 16.83 | 52.54 |
| | | | | | Total | 541.19 |

Table 8: Time to find pairs of near-duplicates by ADA($S$; 100) within $L_\infty \le 0.01$Å between a one database (left) and another (top). The results are symmetric but times are not.

| databases | CSD | | COD | | ICSD | | MP | | GNoME | |
|---|---|---|---|---|---|---|---|---|---|---|
| | count | % | count | % | count | % | count | % | count | % |
| CSD | **4019** | **0.5** | 266761 | 32.1 | 3873 | 0.5 | 0 | 0.0 | 0 | 0.0 |
| COD | 270455 | 78.6 | **11768** | **3.4** | 31135 | 9.1 | 37 | 0.0 | 0 | 0.0 |
| ICSD | 3898 | 3.7 | 32566 | 31.0 | **9606** | **9.1** | 146 | 0.1 | 3 | 0.0 |
| MP | 0 | 0.0 | 29 | 0.0 | 83 | 0.1 | **182** | **0.1** | 12 | 0.0 |
| GNoME | 0 | 0.0 | 0 | 0.0 | 3 | 0.0 | 12 | 0.0 | **4406** | **1.1** |

Table 9: Count and percentage of all pure periodic crystals in each database (left) found to have a near-duplicate in other databases (top) by the distance $L_\infty < 10^{-6}$Å on vectors ADA($S$; 100).

Table 14 reports the found pairs of close entries that differ by PDA up to 0.01Å meaning that these structures can be likely matched by perturbing atoms up to $\frac{0.01}{2}$Å on average.

Table 14 was made within 5 hours on AMD Ryzen 5 5600X 6-core RAM 32Gb due to the ultra-fast search for near-duplicates using the hierarchy AMD, PDD, PDD$^{\{2\}}$.

The 2nd row for 0.01Å says that nearly 30% crystals were deposited in the ICSD multiple times with tiny variations. More than 50% of 0.01-close pairs in all databases (except CSD) differ by atomic types. In all similar (dozens of) cases found in the CSD, the curators concluded that these

| databases | CSD | | COD | | ICSD | | MP | | GNoME | |
|---|---|---|---|---|---|---|---|---|---|---|
| | count | % | count | % | count | % | count | % | count | % |
| CSD | **4013** | **0.5** | 266514 | 32.1 | 3863 | 0.5 | 0 | 0.0 | 0 | 0.0 |
| COD | 270205 | 78.5 | **11754** | **3.4** | 31012 | 9.0 | 14 | 0.0 | 0 | 0.0 |
| ICSD | 3888 | 3.7 | 32455 | 30.9 | **9598** | **9.1** | 73 | 0.1 | 0 | 0.0 |
| MP | 0 | 0.0 | 9 | 0.0 | 36 | 0.0 | **10** | **0.0** | 4 | 0.00 |
| GNoME | 0 | 0.0 | 0 | 0.0 | 0 | 0.0 | 4 | 0.0 | **3248** | **0.8** |

Table 10: Count and percentage of all pure periodic crystals in each database (left) found to have a near-duplicate in other databases (top) by the distance $L_\infty < 10^{-10}$Å on vectors $\mathrm{ADA}(S; 100)$.

| | CSD | COD | ICSD | MP | GNoME | time (min:sec) |
|---|---|---|---|---|---|---|
| CSD | 226.11 | 176.47 | 27.20 | 29.93 | 12.98 | 7:53 |
| COD | 140.85 | 63.19 | 12.72 | 12.07 | 9.35 | 3:58 |
| ICSD | 4.04 | 4.35 | 3.75 | 4.28 | 6.18 | 0:23 |
| MP | 6.11 | 7.06 | 8.85 | 6.94 | 9.72 | 0:39 |
| GNoME | 6.20 | 9.46 | 10.06 | 8.78 | 5.44 | 0:40 |
| | | | | | Total | 8:23 |

Table 11: Time in seconds to find matches by $\mathrm{ADA}_{100}$ within $10^{-6}$ Å between a one database of crystals (left) and another (top). Total time to find all pairs is bottom-right, note that results are symmetric but times are not.

| Database | Duplicates | Groups >1 | Largest group | # Unique | % Unique |
|---|---|---|---|---|---|
| CSD | 36269 | 14656 | 406 | 809513 | 97.40% |
| COD | 30786 | 10536 | 1001 | 323877 | 94.12% |
| ICSD | 42686 | 8081 | 2606 | 70557 | 67.09% |
| MP | 28806 | 4610 | 5362 | 129039 | 84.21% |
| GNoME | 197340 | 33442 | 5607 | 221040 | 57.42% |

Table 12: Information about duplicates within five databases, by $\mathrm{ADA}_{100}$ within 0.01 Å. From left to right: number of entries with a duplicate, number of groups of duplicates, size of the largest group, total number of unqiue structures, percentage of the database which is unique.

| Database | Duplicates | Groups >1 | Largest group | # Unique | % Unique |
|---|---|---|---|---|---|
| CSD | 4013 | 1998 | 5 | 829111 | 99.76% |
| COD | 11754 | 5725 | 9 | 338098 | 98.25% |
| ICSD | 9598 | 3900 | 21 | 99464 | 94.58% |
| MP | 10 | 5 | 2 | 153230 | 100.00% |
| GNoME | 3248 | 1567 | 9 | 383257 | 99.56% |

Table 13: Information about duplicates within five databases, by $\mathrm{ADA}_{100}$ within $10^{-10}$ Å. From left to right: number of entries with a duplicate, number of groups of duplicates, size of the largest group, total number of unique structures, percentage of the database which is unique.

geometric coincidences with different elements are physically impossible, so several journals started investigating the relevant publications. The MP and GNoME consist of simulated crystals obtained by atomic replacements and energy optimization from experimental crystals in the ICSD. The last two rows in Table 14 imply that replacing atoms is easier than genuinely changing crystal geometry.

The CSD and ICSD had a surprisingly large overlap; many of these duplicates are known to the CSD and are intentionally in both databases. Since COD contains both organic and inorganic structures, several thousand matches were found with the ICSD. Out of 35,051 pairs of structures whose geometry matched, 31,918 had the same composition. The others are simple structures where geometry can be identical by coincidence, generally cubic structures with one symmetrically unique site. The Materials Project had few matches with any other database; this is explained by the fact that the geometry of all structures in the Materials Project are changed in the curation process and hence won't match identically even if two crystals are from the same publication, as quoted from

Table 14: Each database has thousands of (near-)duplicates whose all atomic positions can be matched by tiny perturbations. This duplication is unexpected for different compositions as replacing an atom with a different one should stronger affect the geometry.

| near-duplicates | database | $10^{-2}$Å | $10^{-3}$Å | $10^{-4}$Å |
|---|---|---|---|---|
| pairs of | CSD | 8608 | 2403 | 2076 |
| entries | COD | 46646 | 10151 | 6984 |
| within a | ICSD | 291268 | 38351 | 11315 |
| threshold by | MP | 346909 | 32793 | 3333 |
| EMD on PDA | GNoME | 93035 | 3568 | 2742 |
| percentage | CSD | 0.91 | 0.56 | 0.49 |
| of all entries | COD | 5.30 | 4.07 | 3.52 |
| in close pairs | ICSD | 29.53 | 15.51 | 10.03 |
| vs the full | MP | 10.32 | 5.61 | 2.70 |
| database | GNoME | 16.80 | 1.55 | 1.26 |
| percentage | CSD | 0.48 | 0.25 | 0.25 |
| of close pairs | COD | 50.03 | 23.86 | 8.99 |
| with different | ICSD | 78.73 | 63.41 | 56.18 |
| chemical | MP | 99.94 | 99.90 | 99.19 |
| compositions | GNoME | 95.02 | 48.99 | 43.95 |

their documentation page: "We relax all cell and atomic positions in our calculation two times in consecutive runs."

| database | matches | same composition |
|---|---|---|
| CSD | 2,036 | 2,031 |
| COD | 6,435 | 5,893 |
| ICSD | 9,941 | 4,149 |
| MP | 6 | 2 |

Table 15: Number of exact duplicates (PDM within $10^{-10}$Å) in four databases.

Several thousand exact duplicates were found in the CSD, COD and ICSD, some of which are known intentional duplicates. Of the 2,036 duplicates in the CSD, the 5 with different compositions have been previously reported to the CSD prompting investigation. The duplicates with different compositions in the COD and ICSD are simple inorganic cubic structures which can match by coincidence. The relatively few duplicates in the Materials Project database is again explained by their curation process changing the geometry of structures.

The full tables of matches between and duplicates within all databases can be found in the supplementary materials. The tables of duplicates contain all pairs of structures within a tolerance of 0.01Å, and hence have more matches than the reported in Table 15 above. 100 exact duplicates within the CSD, COD and ICSD were further compared with by Earth Mover's distance on the stronger invariant $\text{PDD}^{\{h\}}$ with $h = 2$, confirming they were duplicates. The data in the original database entries of all of these pairs turned out to be the same. Tables of these duplicates can also be found in the supplementary materials.

COMPACK Chisholm & Motherwell (2005) is a heuristic process that tries to overlay molecules of two structures and minimize deviations in atomic positions, as such there is large variability in run time, with some comparisons leaving COMPACK stuck in an infinite loop to eventually time out. It also depends on crystals having well-defined and separate molecules, rather than applying to all periodic point sets. Some pairs of crystals such as the CSD entries HIFCAB and JEPLIA are reported as being distinct by COMPACK despite being geometrically identical. For COMPACK, the median time of 117 ms per comparison is extrapolated to all comparisons in Table 4.

In November 2023, Nature published two papers attracting a lot of interest Peplow (2023):

• Google's DeepMind paper Merchant et al. (2023) claimed that "AI tool GNoME finds 2.2 million new crystals, including 380,000 stable materials that could power future technologies", and

• the Berkeley A-lab paper Szymanski et al. (2023) claimed that " the A-Lab realized 41 novel compounds ... using large-scale ab initio phase-stability data from Materials Project and Google".

Rebutting both papers, domain experts found "scant evidence for compounds that fulfill the trifecta of novelty, credibility, and utility" Cheetham & Seshadri (2024) and concluded that "none of the materials produced by A-lab were new: the large majority were misclassified, and a smaller number were correctly identified but already known" Leeman et al. (2024). Here we additionally review the GNoME database of 384,398 available CIFs goo (2023). The GNoME paper used the Pymatgen structure matcher pym whose first three steps are quoted below:

"1. Given two structures: s1 and s2

2. Optional: Reduce to primitive cells.

3. If the numbers of sites do not match, return False."

If step 2 above is optionally missed, step 3 can output False (no match) for identical crystals given with different non-primitive cells. If step 2 is enforced, step 3 will output False (no match) for any nearly identical crystals, whose primitive cells differ by scaling due to a tiny atomic displacement as in Fig. 1 (right). Since many experimental and simulated structures can differ only slightly, a comparison based on discontinuous properties can miss many near-duplicates.

On another hand, any positive tolerance in all comparisons including Pymatgen (and other software) mathematically leads to all structures being equivalent due to the transitivity axiom. Hence the continuity condition(c) in Problem 1.2 is essential for justified comparisons of crystals.

After filtering all CIFs in GNoME by chemical composition and unit cell volume, we found four CIFs (4135ff7bc7, 6370e8cf86, c6afea2d8e, e1ea534c2c) with equal chemical compositions and unit cell volumes (within $10^{-8}$Å); their CIF files turned out to be identical symbol-for-symbol. In the quadruple 000ce7959c, 5dbe5a510a, f6bf95267d, f6f12f1f29, all atomic coordinates are identical but unit cell parameters differ only in the 6th decimal place in Angstroms.

Further, GNoME contains 68 triples and 1367 pairs of CIFs with equal compositions and cell volumes. Among them, 43 triples and 1089 pairs of CIFs are identical texts, see tables in the supplementary materials. We also found 30K+ CIFs that have identical unit cells (with all parameters to the last digit) to another CIF in GNoME, e.g. two groups of 38 and 39 CIFs with the same unit cells.

The above analysis didn't require any invariants, only comparisons of geometric data in the given CIFs without any transformation by rigid motion. In the past, coincidences in different CIFs were caught manually, e.g. some identical CIFs in GNoME can be found after ordering all CIFs by file size in bytes.

Crystallography experienced several crises in structure determination from the unexpected form of ritonavir Morissette et al. (2003) costing the pharmaceutical industry billions of dollars to the mill of 800 papers Else (2022), which put under investigation nearly 1000 structures in the CSD. These cases can grow in scale by the algorithm below.

**Algorithm A.1** (adversarial generation). *One can generate any number of 'new' structures as follows.*

*1. Take a Crystallographic Information File of a real periodic material from any public database.*

*2. Change a unit cell by applying any integer matrix with determinant 1 to a given basis.*

*3. Arbitrarily extend a unit cell by a random integer factor in each direction of the basis.*

*4. Randomly perturb any cell parameters and atomic coordinates up to a small threshold.*

*5. Replace some non-common chemical elements with similar ones in the periodic table.*

Step 1 can choose a non-famous crystal with at least one non-organic element for a future substitution. Steps 2 and 3 are optional but include many choices to generate more structures. Step 4 is essential because most comparisons miss near-duplicates as in Fig. 1 (right). Step 5 is the final

disguise to avoid detection by chemical composition. One can also check if a new composition has not appeared in the main databases. After obtaining a new CIF (or millions of CIFs), can we deposit these 'new' materials and publish a paper?

On the experimental side, some journals and databases now require extra data such as structure factors from diffraction, which can be perturbed similar to a CIF.

On the computational side, most simulations claim that their materials are 'stable' meaning that their energy is below the convex hull over a spaces of compositions. Hohenberg and Kohn Hohenberg & Kohn (1964) proved the existence of a universal energy potential but there is no explicit formula. Since many algorithms calculate different energies, 'stable' materials are user-dependent.

Even if we fix one easily computable energy such as the Lennard-Jones potential Jones (1924), numerical approximations can slightly deviate from a local energy minimum. Hence adding noise to a real structure might produce near-duplicates that have smaller energies, especially if millions of dollars can buy longer simulations. Geometrically, sampling many points around a vertex on the boundary of a convex hull will likely produce many new vertices on the boundary of a perturbed convex hull. If it is still unclear that Algorithm A.1 can generate millions of *plausibly looking* 'new' materials, we will provide a public implementation. Luckily, mathematics came to the rescue with the counter-algorithm below, which is now being implemented by the Cambridge Crystallographic Data Centre for validating any new structures deposited to the CSD.

**Algorithm A.2** (fast detection of near-duplicate periodic structures). *We find all pairs of periodic structures that differ by atomic displacements up to a given threshold $\varepsilon$.*

*1(a). Split a given database of CIFs into groups with equal (or $\varepsilon$-close) unit cell volumes.*

*1(b). Split each group into subgroups of CIFs with equal (or $\varepsilon$-close) unit cell parameters.*

*1(c). Split each subgroup into subgroups of CIFs whose motifs are $\varepsilon$-close as sets of unordered points. Exclude all the found (near-)duplicates from further comparisons.*

*2. For remaining periodic point sets $S$, compute $\mathrm{PDD}(S; k)$ and $\mathrm{AMD}(S; k)$, say for $k = 100$.*

*3. Find all pairs of structures with distances $L_\infty \leq 2\varepsilon$ between their AMD vectors, which can be done in near-linear time by fast nearest neighbor search Elkin & Kurlin (2023). If $L_\infty > 2\varepsilon$, the structures cannot be obtained from each other by perturbing all atoms up to $\varepsilon$.*

*4. For any remaining pair, compute the Eearth Mover's Distance (EMD) between PDDs, then between $\mathrm{PDD}^{\{2\}}$s. If $\mathrm{EMD} > 2\varepsilon$, the structures cannot be obtained by perturbing all atoms up to $\varepsilon$ due to Theorem 4.1.*

*5. The EMD calculation finds an optimal matching between atoms, so we can check the displacement of any atom to estimate how much structures differ by atomic positions.*

Step 1 is optional and can save time by filtering out easy duplicates, so all thresholds are not essential. Chemists in certain areas can agree not to distinguish materials if atomic displacements are within 0.1Å or 0.01Å. Instead of the angle between basis vectors $v_1, v_2$, Step 1(b) can use the length $|v_1 - v_2|$ of the diagonal for comparisons. Step 1(c) can compare finite sets of unordered points by SCD invariants Widdowson & Kurlin (2023): if $\mathrm{EMD} > 2\varepsilon$ between SCDs, the sets cannot be obtained from each other by perturbing all points up to $\varepsilon$. Algorithm A.2 uses only geometry without chemical elements to counter-act Step 5 in Algorithm A.1 and finds all pairs of structures that can be potentially obtained by atomic displacements up to $\varepsilon$. All other pairs are filtered out due to the Lipschitz continuity of PDD and SCD. The final list of (near-)duplicates might be short enough for traditional chemistry-based validation.

The International Union of Crystallography (IUCr) still discusses changes to the definition of a crystal Brock (2021) because the fundamental question "same or different" has never been rigorously answered. This question was openly asked only in 2020 Sacchi et al. (2020) when the experimental comparisons by the classical tools such as powder diffractions confirmed the unresolved ambiguities that were known since 1944 Patterson (1944).

The IUCr online dictionary iso says that "crystals are said to be isostructural if they have the same structure, but not necessarily the same cell dimensions nor the same chemical composition, and with a 'comparable' variability in the atomic coordinates to that of the cell dimensions and chemical

composition ... CaCO3, NaNO3, and FeBO3 are isostructural." Now a crystal structure is defined as a class of periodic sets under rigid motion in $\mathbb{R}^3$ without repeating the same word "structure", especially because CaCO3, NaNO3, FeBO3 can be geometrically distinguished.

Despite the steady progress in experimental methods Patterson (1944) and mathematical theory Rosenblatt & Seymour (1982), the question "same or different" Sacchi et al. (2020) remained open for homometric structures since 1930 Pauling & Shappell (1930). While the past PDD invariants cannot distinguish infinitely many homometric crystals, the new higher-order $\mathrm{PDD}^{\{h\}}$ distinguished all (infinitely many) such structures in dimensions $n = 1, 2, 3$ by using only order $h = 2$, see Theorem 4.3, Examples 3.3, and 3.6.

# B   APPENDIX: DETAILED PROOFS OF ALL RESULTS

This appendix finishes Example 3.3, illustrates Theorem 4.4 and proves all theorems from section 4.

**Example B.1** (detailed argument why $\mathrm{PDD}^{\{2\}}$ distinguishes $S, Q$ in Example 2.5). *After considering the degenerate cases $c = 0$ and $b \in \{0, 1, 2\}$ in Example 3.3, without loss of generality, we can assume that $1 < b < 2$, then $d_2 > d_3$, $d_5 > \max\{d_4, d_6\}$, $\min\{d_7, d_8, d_9\} > d_6$.*

*The set $S$ in Fig. 3 has a motif of 6 points, which generate isometric triangles $\triangle ABC \simeq \triangle A'B'C'$ with the perimeter $d_2 + d_4 + d_6$, see details in Example 2.5. The other potentially smaller perimeters of triangles on points of $S$ are $d_3 + d_5 + d_6$, $d_3 + d_4 + d_7$. The smallest perimeter for $S$ is the minimum of these sums. The smallest perimeter for $Q$ is the minimum of $d_2 + d_4 + d_5$, $d_2 + d_5 + d_6$, $d_3 + d_4 + d_6$.*

*If $t = d_2 + d_4 + d_6$ equals one of the last sums, one of the following cases holds. if $d_2 = d_3$ then $b = 1$, if $d_4 = d_5$ then $c = 0$, if $d_6 = d_7$ then $b = 2$ (or $b = 0$), so $S \simeq Q$.*

*If $t = d_3 + d_5 + d_6$ is a minimal perimeter for $S$, then $t$ can't equal any of the three sums for $Q$. Indeed, if $t = d_2 + d_5 + d_6$ then $d_2 = d_3$. If $t = d_3 + d_4 + d_6$ then $d_4 = d_5$. The minimality of $t$ for $S$ means that $d_3 + d_6 < d_2 + d_4$, so $t = d_3 + d_5 + d_6$ can't equal $d_2 + d_4 + d_5$ for $Q$.*

*If $t = d_3 + d_4 + d_7$ is a minimal perimeter for $S$, then $t$ can't equal any of the three sums for $Q$. Indeed, if $t = d_3 + d_4 + d_6$ then $d_6 = d_7$. The minimality of $t$ for $S$ means that $d_3 + d_7 < d_2 + d_6 < d_2 + d_5$, so $t = d_3 + d_4 + d_7 < d_2 + d_4 + d_5$ for $Q$. Similarly, $d_4 + d_7 < d_5 + d_6$ implies that $t = d_3 + d_4 + d_7 < d_3 + d_5 + d_6 < d_2 + d_5 + d_6$.*

*In all these cases, $S, Q$ become isometric. Hence the smallest perimeters in $\mathrm{PDD}^{\{2\}}$ for $k = 1$ distinguish all pairs of the homometric sets $S, Q$. The same conclusion holds for more general sets obtained from $S, Q$ by periodic translations in other directions (along the $y$-axis or even in any $\mathbb{R}^n$), see Fig. 10 in Pozdnyakov & Ceriotti (2022), when extra periods are large and don't affect any triangles with the smallest perimeters.*

Since any lattice $\Lambda \subset \mathbb{R}^n$ has a single point in a motif, any Pointwise Distance Distribution $\mathrm{PDD}^{\{h\}}(\Lambda; k)$ is a single row of the length $k$, which can be visualized as a polygonal curve depending on $k$. Fig. 15 illustrates Theorem 4.4 for $h = 2, 3$ and six basic lattices $\Lambda \subset \mathbb{R}^2$, and supports the conjecture that $\dfrac{a(h, k)}{\sqrt[n]{b(h, k)}}$ has a limit as $k \to +\infty$ for any order $h > 1$.

Fig. 16 shows the six 2D lattices illustrating the asymptotic behaviour of $\mathrm{PDD}^{\{h\}}$ in Fig. 15.

An explicit upper bound for the time complexity in Theorem 4.5 will be proved after Theorem 4.4, because both results will use Lemmas B.5, B.6, B.7. The proof of Theorem 4.1 is split into the parts (PDD and PDM) based on Lemmas B.3 and B.4, respectively. We start from Theorem B.2, which proves the invariance of $\mathrm{PDD}^{\{h\}}$ under isometry and changes of a unit cell, and can be considered a partial case of Theorem 4.1 for perturbation $\varepsilon = 0$.

**Theorem B.2** (invariance of $\mathrm{PDD}^{\{h\}}$). *For a finite unordered set $S$ in any metric space or a periodic point set $S$ in any $\mathbb{R}^n$, the higher-order Pointwise Distance Distribution $\mathrm{PDD}^{\{h\}}(S; k_1, \ldots, k_h)$ from Definition 3.1 is an isometry invariant of the set $S$ for any integers $h, k_1, \ldots, k_h \geq 1$.*

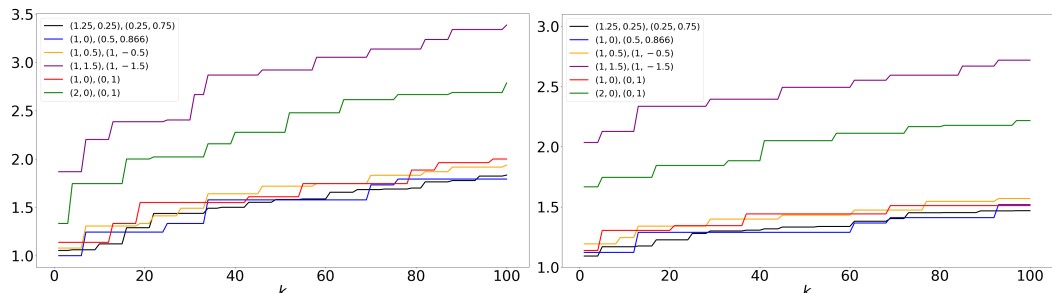

Figure 15: The asymptotic behaviour of the higher-order $\text{PDD}^{\{2\}}(\Lambda; k)$ and $\text{PDD}^{(3)}(\Lambda; k)$ for the six lattices $\Lambda \subset \mathbb{R}^2$ in Fig. 16. **Left**: $h = 2$. **Right**: $h = 3$.

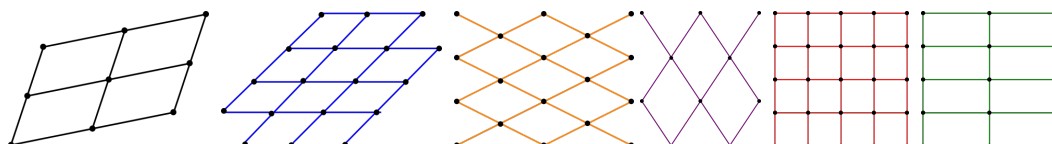

Figure 16: The 2D lattices used in Fig. 15. **1st**: a generic black lattice $\Lambda_1$ with the basis $(1.25, 0.25), (0.25, 0.75)$ and $c(\Lambda_1) = \sqrt{\dfrac{7}{8\pi}} \approx 0.525$. **2nd**: the blue hexagonal lattice $\Lambda_2$ with the basis $(1, 0), (1/2, \sqrt{3}/2)$ and $c(\Lambda_2) = \sqrt{\dfrac{\sqrt{3}}{2\pi}} \approx 0.528$. **3rd**: the orange rhombic lattice $\Lambda_3$ with the basis $(1, 0.5), (1, -0.5)$ and $c(\Lambda_3) = \sqrt{\dfrac{1}{\pi}} \approx 0.564$. **4th**: the purple rhombic lattice $\Lambda_4$ with the basis $(1, 1.5), (1, -1.5)$ and $c(\Lambda_4) = \sqrt{\dfrac{3}{\pi}} \approx 0.977$. **5th**: the red square lattice $\Lambda_5$ with the basis $(1, 0), (0, 1)$ and $c(\Lambda_5) = \sqrt{\dfrac{1}{\pi}} \approx 0.564$. **6th**: the green rectangular lattice $\Lambda_6$ with the basis $(2, 0), (0, 1)$ and $c(\Lambda_6) = \sqrt{\dfrac{2}{\pi}} \approx 0.798$.

For a finite set $S$ of $m$ points, if any $k_i$ is greater than the number $\binom{m-1}{i}$ of $(i + 1)$-tuples with a fixed point $p \in S$, we set all superfluous sums to the last maximum value.

***Proof of Theorem B.2.*** Firstly, for any periodic point set $S \subset \mathbb{R}^n$, we show that scaling up a unit cell $U$ to a non-primitive cell keeps $\text{PDD}^{\{h\}}$ invariant. It suffices to scale up $U$ by a factor $l$, say along the first basis vector $v_1$ of $U$, then the number $m$ of motif points of $S$ is multiplied by $l$.

Then the matrix $D_{lU}(S; k_1, \ldots, k_h)$ consisting of smallest average sums in Definition 3.1 has the larger size $lm \times (\sum\limits_{i=1}^{h} k_i)$ in comparison with the original $m \times (\sum\limits_{i=1}^{h} k_i)$ matrix $D_U(S; k_1, \ldots, k_h)$ but each row is repeated $l$ times for the shifted points $p + iv_1$, where $p$ is any point from the original motif $M = S \cap U$ of $S$, for $i = 0, \ldots, l - 1$.

Secondly, we show that the matrix $D_U(S; k_1, \ldots, k_h)$, hence $\text{PDD}(S; k_1, \ldots, k_h)$, is independent of a primitive cell $U$. Let $U, V$ be any primitive cells of a periodic set $S \subset \mathbb{R}^n$ with a lattice $\Lambda$. Any point $q \in S \cap V$ can be translated by a vector of $\Lambda$ to a point $p \in S \cap U$ and vice versa. These translations preserve distances and establish a bijection between the motifs $S \cap U \leftrightarrow S \cap V$, and a bijection between all rows of $D_U(S; k_1, \ldots, k_h) \leftrightarrow D_V(S; k_1, \ldots, k_h)$.

Thirdly, we prove that $\text{PDD}^{\{h\}}(S; k_1, \ldots, k_h)$ is preserved by any isometry $f : S \to Q$. Any primitive cell $U$ of $S$ is bijectively mapped by $f$ to the unit cell $f(U)$ of $Q$, which should be also primitive. Indeed, if $Q$ is preserved by a translation along a vector $v$ that doesn't have all integer coefficients in the basis of $f(U)$, then $S = f^{-1}(Q)$ is preserved by the translation along $f^{-1}(v)$,

which doesn't have all integer coefficients in the basis of $U$, so $U$ was non-primitive. Since $U$ and $f(U)$ have the same number of points from $S$ and $Q = f(S)$, the isometry $f$ gives a bijection between the motifs $S \cap U \leftrightarrow Q \cap f(U)$.

For any finite or periodic sets $S, Q$, since $f$ maintains distances, the $k$ smallest average sums of all pairwise distances between any point $p \in S \cap U$ and $p_1, \ldots, p_h \in S$, equal the same sums for $f(p) \in Q \cap f(U)$ and $f(p_1), \ldots, f(p_h) \in Q$. These coincidences of all sums imply that $\text{PDD}^{\{h\}}(S; k_1, \ldots, k_h) = \text{PDD}^{\{h\}}(Q; k_1, \ldots, k_h)$ up to a permutation of rows. $\qquad\square$

Recall that the distance $L_\infty$ between ordered lists of $k$ real numbers (or vectors $A = (a_1, \ldots, a_k)$ and $B = (b_1, \ldots, b_k)$ in $\mathbb{R}^k$) is $L_\infty(A, B) = \max_{i=1,\ldots,k} |a_i - b_i|$.

**Lemma B.3** (perturbation of an ordered list). *Let $0 \leq a_1 \leq \cdots \leq a_k$ be a list $A$ of ordered real numbers. For some $\varepsilon \geq 0$, let a map $g$ perturb each $a_i$ to $g(a_i)$ so that $|g(a_i) - a_i| \leq \varepsilon$ for $i = 1, \ldots, k$. Let $B$ be the list obtained by putting $g(a_1), \ldots, g(a_k)$ in increasing order. Then $L_\infty(A, B) \leq \varepsilon$.*

*Proof.* It suffices to prove that the $i$-th number $b_i = g(a_j)$ in the ordered list $B$ is $\varepsilon$-close to the $i$-th number $a_i$ in the original list $A$, so $a_i - \varepsilon \leq b_i \leq a_i + \varepsilon$ for $i = 1, \ldots, k$. Firstly, assume by contradiction that $b_i < a_i - \varepsilon$.

Since every number of $A$ was perturbed by at most $\varepsilon$, the $i$ numbers $b_1 \leq \cdots \leq b_i < a_i - \varepsilon$ can be obtained only as perturbations of numbers from $A$ that are strictly less than $a_i$. However, the ordered list $A$ has at most $i - 1$ numbers that are less $a_i$. This contradiction proves that $b_i \geq a_i - \varepsilon$. The similar argument proves that $b_i \leq a_i + \varepsilon$. $\qquad\square$

***Proof of Theorem** 4.1 for* $\text{PDD}^{\{h\}}$. Let a map $g$ perturb any point $p \in S$ to an $\varepsilon$-close point $g(p) \in Q$ so that $d(g(p), p) \leq \varepsilon$. Here $d$ can denote a base metric (if $S$ is finite) in a metric space containing $S$ or the Euclidean distance in the case of a periodic set $S \subset \mathbb{R}^n$.

In the periodic case, if the perturbation is small enough so that $\varepsilon < r(S)$, Lemma 7 from Widdowson et al. (2022) proves that $S, Q$ have a common lattice with a unit cell $U$ such that $S = \Lambda + (S \cap U)$ and $Q = \Lambda + (Q \cap U)$. Then $S, Q$ share a unit cell $U$ and have the same number $m = m(S) = m(Q)$ of points in $U$. Expand $\text{PDD}^{\{h\}}$ of both $S, Q$ to the matrices with $m$ equally weighted rows. Reorder $m$ rows of these matrices according to the bijection $p \mapsto g(p)$ for $p \in S \cap U$.

Since each point $p \in S$ is perturbed up to $\varepsilon$, any distance $d(p, q)$ between $p, q \in S$, hence any average sum $a$ from Definition 3.1, changes by at most $2\varepsilon$ due to the triangle inequality for the metric $d$. Recall that by Definition 3.1 the $m \times (\sum_{i=1}^{h} k_i)$ matrix $D(S; k_1, \ldots, k_h)$ is considered a concatenation of the $h$ smaller $m \times k_j$ matrices $D^{\{j\}}(S; k_j)$, one for every order $j = 1, \ldots, h$.

Some of the average sums from each original matrix $D^{\{j\}}(S; k_j)$ can increase up to $2\varepsilon$ and will be outside the $k_j$ smallest average sums in the new matrix $D^{\{j\}}(S; k_j)$ for $i = 1, \ldots, h$. In this case, for each row $i = 1, \ldots, m$ and $j = 1, \ldots, h$, let $k(i, j) \geq k_j$ be the maximum index such that the $k(i, j)$-th smallest average sum (of pairwise distances between $j + 1$ points including $p_i \in S$) for $S$ is at most $2\varepsilon$ plus the largest average sum on $j + 1$ points from the original matrix $D^{\{j\}}(S; k_j)$ in the $i$-th row.

Set $k'_j = \max_{i=1,\ldots,m} k(i, j) \geq k_j$ for $j = 1, \ldots, h$. Then the $i$-th row of $D^{\{j\}}(Q; k_j)$ is obtained from the $i$-th row of $D^{\{j\}}(S; k'_j)$ of the length $k'_j$ by changing every value by at most $2\varepsilon$, putting them in increasing order, and taking only the first $k_j \leq k'_j$ smallest values. For each $i = 1, \ldots, m$ and $j = 1, \ldots, h$, Lemma B.3 implies that the $i$-th rows of the extended length $k'_j$ differ in $D^{\{j\}}(S; k'_j)$ and its $2\varepsilon$-perturbation by at most $2\varepsilon$ in the metric $L_\infty$.

The same conclusion holds for the shorter $i$-th rows $R_{i,j}(S)$ and $R_{i,j}(Q)$ of the original length $k_j$ in the matrices $D^{\{j\}}(S; k_j)$ and $D^{\{j\}}(Q; k_j)$, respectively, so $L_\infty(R_{i,j}(S), R_{i,j}(Q)) \leq 2\varepsilon$. For each

of $S, Q$, concatenate the $h$ rows $R_{i,1}, \ldots, R_{i,h}$ into one row $R_i$ of the length $\sum_{i=1}^{h} k_i$, which maintains the same upper bound $L_{\infty}(R_i(S), R_i(Q)) \leq 2\varepsilon$ for $i = 1 \ldots, m$.

To prove that EMD $\leq 2\varepsilon$, define the simple 1-1 partial flows from the $m$ rows of $D(S; k_1, \ldots, k_h)$ to the $m$ rows of $D(Q; k_1, \ldots, k_h)$ by setting $f_{ii} = \dfrac{1}{m}$ and $f_{ij} = 0$ for $i \neq j$, where $i, j = 1, \ldots, m$. Then

$$\text{EMD}(\text{PDD}(S; k_1, \ldots, k_h), \text{PDD}(Q; k_1, \ldots, k_h)) \leq$$

$$\sum_{i,j=1}^{m} f_{ij} L_{\infty}(R_i(S), R_j(Q)) = \frac{1}{m} \sum_{i=1}^{m} L_{\infty}(R_i(S), R_i(Q))$$

$\leq 2\varepsilon$ since EMD minimizes the cost over all choices of $f_{ij}$ subject to the constraints of Definition 3.5. $\qquad\square$

The second part of Theorem 4.1 for PDM needs Lemma B.4 estimating derivatives of moments.

**Lemma B.4** (derivatives of moments). *For any vector $A = (a_1, \ldots, a_m)$ of positive real numbers, the $l$-th moment $\mu_l(A) = \sqrt[l]{m^{1-l} \sum_{i=1}^{m} w_i a_i^l}$ with fixed weights $w_1, \ldots, w_m > 0$ such that $\sum_{i=1}^{m} w_i = 1$ has $\sum_{i=1}^{m} \dfrac{\partial \mu_l}{\partial a_i} \leq 1$.*

*Proof.* For simplicity, we first remove the factor $m^{(1/l)-1}$.

$$\frac{\partial \mu_l}{\partial a_i} = w_i a_i^{l-1} \left( \sum_{i=1}^{m} w_i a_i^l \right)^{(1/l)-1} \leq w_i^{1/l} \left( \frac{w_i a_i^l}{\sum_{i=1}^{m} w_i a_i^l} \right)^{(l-1)/l}$$

$\leq w_i^{1/l}$, where we used $w_i a_i^l \leq \sum_{i=1}^{m} w_i a_i^l$ for $a_1, \ldots, a_m > 0$. The power means inequality in section 3.1 of Bullen (2003) implies that

$$\frac{1}{m} \sum_{i=1}^{m} w_i^{1/l} \leq \left( \frac{1}{m} \sum_{i=1}^{m} w_i \right)^{1/l} = m^{-1/l},$$

so $\sum_{i=1}^{m} w_i^{1/l} \leq m^{1-(1/l)}$. After reinstating the factor $m^{(1/l)-1}$, we get $\sum_{i=1}^{m} \dfrac{\partial \mu_l}{\partial a_i} \leq m^{(1/l)-1} \sum_{i=1}^{m} w_i^{1/l} \leq 1$. $\qquad\square$

***Proof of Theorem*** *4.1 for* PDM. To prove the Lipschitz continuity of the $l \times (\sum_{i=1}^{h} k_i)$ matrix PDM$[l](S; k_1, \ldots, k_h)$, take any column $A = (a_1, \ldots, a_m)$ of PDD$(S; k_1, \ldots, k_h)$. Due to the proved continuity of $A$ in the metric $L_{\infty}$ with the Lipschitz constant $\lambda = 2$, it suffices to check that $|\mu_l(B) - \mu_l(A)| \leq L_{\infty}(A, B)$ for any $l$-th moment $\mu_l(A) = \sqrt[l]{m^{1-l} \sum_{i=1}^{m} w_i a_i^l}$ and any vectors $A, B \in \mathbb{R}^m$.

Consider the function $f_l(t) = \mu_l(tB + (1-t)A) - \mu_l(A)$ for $t \in [0, 1]$, so $f_l(0) = 0$ and $f_l(1) = \mu_l(B) - \mu_l(A)$. Mean value Theorem 5.10 in Rudin et al. (1976) says that $f_l(1) - f_l(0) = \dfrac{df_l}{dt}(t_0) \cdot$

$(1 - 0)$ for some $t_0 \in [0, 1]$, so $0 \leq f_l(1) \leq \max\limits_{0 \leq t \leq 1} \left| \dfrac{df_l}{dt} \right|$. It remains to bound the derivative:

$$\left| \frac{df_l}{dt} \right| = \left| \sum_{i=1}^{m} \frac{\partial \mu_l}{\partial a_i} \cdot (b_i - a_i) \right| \leq \max_{i=1,\ldots,m} |a_i - b_i| \sum_{i=1}^{m} \frac{\partial \mu_l}{\partial a_i}$$

$\leq \max\limits_{i=1,\ldots,m} |a_i - b_i| = L_\infty(A, B)$ by Lemma B.4. $\qquad \square$

***Proof of Theorem 4.3.*** For a finite set $S \subset \mathbb{R}$ of $m$ unordered points, we prove that $S$ can be reconstructed from $\mathrm{PDD}(S; m-1)$ uniquely up to isometry. Indeed, the number $m$ can be assumed to be known as one plus the number columns in $\mathrm{PDD}(S; m-1)$. Find a row $R$ whose last distance $d$ is maximal across the whole $\mathrm{PDD}(S; m-1)$.

This maximal distance is achieved exactly for two most distant points of $S$, otherwise $\mathrm{PDD}(S; m-1)$ is unrealizable by $S$. These two most distant points can be fixed at the positions $0$ and $d$ up to isometry of $\mathbb{R}$. All other $m - 2$ points of $S$ are uniquely determined by the first $m - 2$ distances in the row $R$, which should be all distinct.

For a periodic sequence $S \subset \mathbb{R}$, the Pointwise Shift Distribution $\mathrm{PSD}(S; k)$ similarly to PDD whose rows are unordered as for $\mathrm{PDD}(S; k)$, is invariant under rigid motion, which is a translation in $\mathbb{R}$. Hence EMD is a metric on PSDs, which we consider weighted distributions of unordered rows. The Lipschitz continuity of $\mathrm{PSD}(S; k)$ is almost identical to Theorem 4.1 for $\mathrm{PDD}(S; k)$.

The time to compute $\mathrm{PDD}(S; k)$ is quadratic in the size $m$ of a motif and linear in the number $k$ of neighbors. Indeed, $S$ have a motif $M$ of $m$ points $0 = p_0 < p_1 < \cdots < p_{m-1} < p_m$ and period $L = p_m - p_0$. For any point $p_i \in M$, the distance to its $k$-th neighbor is $p_{i+k-mN} - p_i + LN$, where $N = [k/m]$ is the integer part and $p_j = p_{j-m} + L$ for $m \leq j < 2m$. So all $k$ neighbors of $p_i$ are computed in linear time in both $k, m$, hence the total time over $m$ points of $M$ is quadratic in $m$.

Now we prove that any periodic point set $S \subset \mathbb{R}$ can be reconstructed (uniquely up to translation) from any row $a_1 < \cdots < a_{m-1} < a_m$ of $\mathrm{PSD}(S; m)$ by writing the points of a motif as $p_k = a_{k+1} - a_1$ for $k = 0, \ldots, m-1$, where $p_0 = 0$, and setting the period of $S$ to $d_m$.

The number $m$ is given as the number of columns of $\mathrm{PSD}(S; m)$. The completeness can be stated as follows: any periodic sequences $S, Q \subset \mathbb{R}$ whose motifs have at most $m$ points are related by translation if and only if $\mathrm{PSD}(S; m) = \mathrm{PSD}(Q; m)$ as weighted distributions of unordered rows. $\qquad \square$

The invariant $\mathrm{PSD}(S; k)$ can be enhanced to a complete invariant under isometry (including reflections) in $\mathbb{R}$ as follows. Let $\bar{S}$ be the mirror image of $S$ under reflection $x \mapsto -x$. In any row $a_1 < \cdots < a_k$ of $\mathrm{PSD}(S; k)$ for $k \geq m$, we can use the $m$-th distance $a_m$ equal to the period $L$ to write the corresponding row

$$L - a_{m-1} < \cdots < L - a_1 < 2L - a_{m-1} < \ldots$$

in the new matrix $\mathrm{PSD}(\bar{S}; k)$. Any periodic sequences $S, Q$ are related by isometry in $\mathbb{R}$ if and only if $\mathrm{PSD}(S; m) = \mathrm{PSD}(Q; m)$ or $\mathrm{PSD}(\bar{S}; m) = \mathrm{PSD}(Q; m)$.

**Lemma B.5** (bounds of distances and their averages). *Let $S \subset \mathbb{R}^n$ be any periodic point set. For any $h, k \geq 1$ and a point $p \in S$, let $a(h, k)$ be the $k$-th smallest average sum achieved for of all pairwise distances between $p$ and $h$ other points $p_1, \ldots, p_h \in S$, see Definition 3.1. Set $R = \max\limits_{i=1,\ldots,h} |p_i - p|$.*

*Then $\dfrac{2R}{h+1} \leq a(h, k) \leq \dfrac{2hR}{h+1}$.*

*Proof.* After translating $p \in S$ to the origin $0 \in \mathbb{R}^n$, one can assume that $p = 0$. Let $p_1 \in S$ be a point such that $R = |p_1| = \max\limits_{i=1,\ldots,h} |p_i|$. For any other point $p_i \neq p_1$, the triangle inequalities $|p_i| + |p_1 - p_i| \geq |p_1| = R$ imply that

$$a(h, k) = \frac{2}{h(h+1)} \sum_{0 \leq i < j \leq h} |p_i - p_j| \geq$$

$$\geq \frac{2}{h(h+1)} \left( |p_1| + \sum_{i=2}^{h} (|p_i| + |p_1 - p_i|) \right) \geq$$

$$\geq \frac{2}{h(h+1)} \left( R + \sum_{i=2}^{h} R \right) = \frac{2R}{h+1}.$$

For the upper bound of $a(h, k)$, we use $|p_i| \leq R$ and the triangle inequalities $|p_i - p_j| \leq |p_i| + |p_j| \leq 2R$ as follows:

$$a(h, k) = \frac{2}{h(h+1)} \left( \sum_{i=1}^{h} |p_i| + \sum_{1 \leq i < j \leq h} |p_i - p_j| \right) \leq$$

$$\leq \frac{2}{h(h+1)} \left( \sum_{i=1}^{h} R + \sum_{1 \leq i < j \leq h} 2R \right) =$$

$$= \frac{2}{h(h+1)} \left( hR + \frac{h(h-1)}{2} 2R \right) = \frac{2hR}{h+1},$$

which finishes the proof of the upper bound. $\square$

For $h = 1$, the bounds of Lemma B.5 give the exact equality $a(1, k) = R$. Lemma B.6 was proved in a slightly more general form in Lemma 11 from Widdowson et al. (2022).

**Lemma B.6** (number of points in a ball). *Let $S \subset \mathbb{R}^n$ be any periodic point set with a unit cell $U$, which has $m$ points of $S$ and generates a lattice $\Lambda$ and has a longest diagonal $d$. For any point $p \in S \cap U$ and a radius $r$, consider*

$$U_-(p; r) = \bigcup_{v \in \Lambda} \{(U + v) \text{ such that } (U + v) \subset \bar{B}(p; r)\},$$

$$U_+(p; r) = \bigcup_{v \in \Lambda} \{(U + v) \text{ such that } (U + v) \cap \bar{B}(p; r) \neq \emptyset\}.$$

*Then the number of points of $S$ in the closed ball $\bar{B}(p; r)$ with the center $p$ and any radius $r \geq d$ has the bounds* $\left( \frac{r-d}{c(S)} \right)^n \leq m \frac{\text{vol}[U_-(p; r)]}{\text{vol}[U]} \leq |S \cap \bar{B}(p; r)| \leq m \frac{\text{vol}[U_+(p; r)]}{\text{vol}[U]} \leq \left( \frac{r+d}{c(S)} \right)^n,$ *where $c(S) = \sqrt[n]{\frac{\text{vol}[U]}{mV_n}}$, $\text{vol}[U]$ is the volume of $U$, $V_n$ is the unit ball volume.* ∎

For Theorem 4.4, we prove the following slightly updated bounds: $\frac{2}{h+1} \left( c(S) \sqrt[n]{b(h, k)} - d \right) \leq a(h, k) \leq \frac{2h}{h+1} \left( c(S) \sqrt[n]{b(h, k)} + d \right)$ for $k \geq 1$, where $b(h, k)$ equals any real number $b + 1$ such that $b \geq h$ and $\binom{b}{h} = \frac{b(b-1) \ldots (b-h+1)}{h!} \in (k-1, k]$, e.g. one can set $b(1, k) = 1 + k$ and $b(2, k) = 1.5 + \sqrt{2k}$.

**Lemma B.7** (increasing binomial coefficient). *For any fixed integer $h \geq 1$, the binomial coefficient $\binom{b}{h} = \frac{b(b-1) \ldots (b-h+1)}{h!}$ is strictly increasing for any real $b \geq h$ so that if $h \leq b < c$ then $\binom{b}{h} < \binom{c}{h}$.*

*Proof.* The derivative $\frac{d}{dx} \binom{x}{h} > 0$ for any $x \geq h$. $\square$

***Proof of Theorem*** *4.4.* To prove the lower bound for the $k$-th smallest sum $a(h,k)$, set $r = \dfrac{h+1}{2}a(h,k)$. For any point $p$ in a motif of $S$, consider the closed ball $\bar{B}(p;r)$ with the center $p$ and radius $r$. By the lower bound of Lemma B.5, all points $p_1,\ldots,p_h \in S$ that are used for computing $a(h,k)$ have $R = \max\limits_{i=1,\ldots,h}|p_i - p| \le \dfrac{h+1}{2}a(h,k) = r$ and hence belong to the ball $\bar{B}(p;r)$.

By the upper bound of Lemma B.6, if this ball contains $l$ points of $S$ (excluding $p$), then $l+1 \le \left(\dfrac{r+d}{c(S)}\right)^n$. By using one fixed point $p$ and any $h$ of $l$ other distinct points $p_1,\ldots,p_h \in S \cap \bar{B}(p;r)$, we can form $\begin{pmatrix} l \\ h \end{pmatrix} = \dfrac{l(l-1)\ldots(l-h+1)}{h!}$ tuples $p, p_1, \ldots, p_h$ whose average sums of all pairwise distances should include all $k$ smallest values up to the $k$-th $a(h,k)$. Hence $\begin{pmatrix} l \\ h \end{pmatrix} \ge k$.

For $l \ge h = 2$, the last inequality is $\dfrac{l(l-1)}{2} \ge k$, $l^2 - l - 2k \ge 0$, $l \ge \dfrac{1 + \sqrt{1+8k}}{2} \ge 0.5 + \sqrt{2k}$.

For any $h \ge 1$, let $b(h,k) = b+1$ satisfy $b \ge h$ and $\begin{pmatrix} b \\ h \end{pmatrix} = \dfrac{b(b-1)\ldots(b-h+1)}{h!} \in (k-1,k]$, e.g. one can set $b(2,k) = 1.5 + \sqrt{2k}$. By Lemma B.7, $\begin{pmatrix} l \\ h \end{pmatrix} \ge k$ for $l \ge h$ implies that $l \ge b = b(h,k) - 1$. Then

$$\left(\frac{r+d}{c(S)}\right)^n \ge l+1 \ge b(h,k), \quad \frac{r+d}{c(S)} \ge \sqrt[n]{b(h,k)},$$

$$\frac{h+1}{2}a(h,k) = r \ge c(S)\sqrt[n]{b(h,k)} - d,$$

$$a(h,k) \ge \frac{2}{h+1}\Big(c(S)\sqrt[n]{b(h,k)} - d\Big).$$

To prove the upper bound for the $k$-th sum $a(h,k)$, set $R = \dfrac{h+1}{2h}a(h,k)$ and consider any $r < R$.

By the upper bound of Lemma B.5, $p$ with any other $h$ points $p_1,\ldots,p_h \in S \cap \bar{B}(p;r)$ have average sums that are at most $\dfrac{2hr}{h+1} < \dfrac{2hR}{h+1} = a(h,k)$, so less than the $k$-th smallest sum $a(h,k)$. If the ball $\bar{B}(p;r)$ contains $l$ points of $S$ (excluding $p$), then these points can form at most $k-1$ tuples consisting of $p$ and $h$ of $l$ other vertices, so $\begin{pmatrix} l \\ h \end{pmatrix} \le k-1$. By Lemma B.7 for $b = b(h,k) - 1 \ge h$, $\begin{pmatrix} b \\ h \end{pmatrix} = \dfrac{b(b-1)\ldots(b-h+1)}{h!} \in (k-1,k]$ implies that $l < b = b(h,k) - 1$. Lemma B.6 gives us

$$\left(\frac{r-d}{c(S)}\right)^n \le l+1 < b(h,k), \quad \frac{r-d}{c(S)} < \sqrt[n]{b(h,k)}.$$

Since the resulting inequality $r < c(S)\sqrt[n]{b(h,k)} + d$ holds for all $r < R$, where $R = \dfrac{h+1}{2h}a(h,k)$ is fixed, we get

$$\frac{h+1}{2h}a(h,k) = R \le c(S)\sqrt[n]{b(h,k)} + d,$$

$$a(h,k) \le \frac{2h}{h+1}\Big(c(S)\sqrt[n]{b(h,k)} + d\Big).$$

If $h = 1$, both bounds have the same main term:

$$c(S)\sqrt[n]{b(1,k)} - d \le a(h,k) \le c(S)\sqrt[n]{b(1,k)} + d.$$

If we divide both sides by $\sqrt[n]{k}$ and $k \to +\infty$, we get $\lim\limits_{k \to +\infty} \dfrac{a(1,k)}{\sqrt[n]{k}} = c(S)$. We replaced $k+1$

with $k$ in $b(1,k)$ because $\lim\limits_{k \to +\infty} \dfrac{\sqrt[n]{k+1}}{\sqrt[n]{k}} = 1$ for any fixed dimension $n$.

For similar reasons and $h = 2$, the ratio $\dfrac{a(2,k)}{\sqrt[2n]{2k}}$ has the asymptotic bounds $\dfrac{2}{3}c(S)$ and $\dfrac{4}{3}c(S)$ as

$k \to +\infty$. $\qquad\qquad\qquad\qquad\qquad\qquad\qquad\qquad\qquad\qquad\qquad\qquad\qquad\qquad\qquad\quad$ $\square$

We conjecture that $\lim\limits_{k \to +\infty} \dfrac{a(h,k)}{\sqrt[hn]{hk}}$ exists for any $h \geq 2$. If yes and this limit differs from $c(S) =$

$\sqrt[n]{\dfrac{\mathrm{vol}[U]}{mV_n}}$, it can be named the $h$-order *point packing coefficient* $c(S;h)$.

**Corollary B.8** (bounds for distances to neighbors). *Let a periodic point set $S \subset \mathbb{R}^n$ have a unit cell with a longest diagonal $d$. For any point $p \in S$, the distance $a(1,k)$ to its $k$-th nearest neighbor in $S$ has the bounds*

$$c(S)\sqrt[n]{k+1} - d \leq a(1,k) \leq c(S)\sqrt[n]{k+1} + d \text{ for } k \geq 1.$$

*Proof.* Use Theorem 4.4 for $h = 1$, $b(1,k) = \sqrt[n]{k+1}$. $\qquad\qquad\qquad\qquad\qquad\qquad\qquad$ $\square$

**Lemma B.9** (upper bound of a binom). *For any integer $n \geq 1$ and real $a, b \geq 0$, we have $(a+b)^n \leq 2^n(a^n + b^n)$.*

*Proof.* Due to $a^i b^{n-i} \leq (\max\{a,b\})^n \leq a^n + b^n$, the binomial formula gives $(a+b)^n = \sum\limits_{i=0}^{n} \binom{n}{i} a^i b^{n-i} \leq (a^n + b^n) \sum\limits_{i=0}^{n} \binom{n}{i} = 2^n(a^n + b^n)$. $\qquad\qquad\qquad\qquad\quad$ $\square$

Theorem 4.5 will be proved in the following explicit form. Let a periodic set $S \subset \mathbb{R}^n$ have $m$ points in a unit cell $U$ whose longest diagonal has a length $d$. Recall that $V_n$ is the unit ball volume in $\mathbb{R}^n$ and introduce the *skewness* $\nu(U) = \dfrac{d}{\sqrt[n]{\mathrm{vol}[U]}}$ of the cell $U$. For any $h, k \geq 1$, the number of operations to compute $\mathrm{PDD}^{\{h\}}(S;k)$ will be proved to be proportional to at most $mN \log N$, where

$$N \leq \frac{2^h}{h!}(2h+3)^{hn}\left((2h+3)^h k + (V_n \nu(U)m)^{hn}\right).$$

If $h = 1$, the simpler estimate will be $N \leq 2^n(k+2) + (5V_n\nu(U)m)^n$. The time $mN \log N$ is near-linear in the number $k$ of neighbors and polynomial of degree $hn + 1$ in the motif size $m$ (with logarithmic factors) for any $h \geq 1$.

***Proof of Theorem*** *4.5.* Let the origin $0 \in \mathbb{R}^n$ be at the center of the unit cell $U$. If $d$ is the length of a longest diagonal of $U$, then any point $p \in M = S \cap U$ is covered by the closed ball $\bar{B}(0, 0.5d)$. By Corollary B.8, the distance $a(1,k)$ from any point $p \in M$ to its $k$-th nearest neighbor in $S$ has the upper bound $a(1,k) \leq c(S)\sqrt[n]{k+1} + d$. Then all $k$ neighbors of $p$ in $S$ are covered by the single ball $\bar{B}(0; r(1,k))$ of the radius $r(1,k) = c(S)\sqrt[n]{k+1} + 1.5d$.

For a fixed point $p$ and any $h > 1$, to find a similar ball including all points that are needed to compute the $k$ smallest average sums $a(h,1) \leq \cdots \leq a(h,k)$, we start from the integer number $l = \lceil b(h,k) - 1 \rceil$ of closest neighbors $p_1, \ldots, p_l$ of $p$, where $b(h,k)$ is any real $b+1$ such that $b \geq h$ and $\binom{b}{h} \in (k-1, k]$. Then $\binom{l}{h} \geq k$ by Lemma B.7. Since the $l+1$ points $p, p_1, \ldots, p_l$ are covered by the ball $\bar{B}(p; R)$ of the radius $R = \max\limits_{i=1,\ldots,l} |p_i - p|$, the lower bound of Lemma B.6 gives

$$\left(\frac{R-d}{c(S)}\right)^n \leq l + 1 \leq b(h,k) + 1, \text{ so } R \leq c(S)\sqrt[n]{b(h,k)+1} + d.$$

All $\begin{pmatrix} l \\ h \end{pmatrix} \geq k$ average sums of pairwise distances between $p$ and any $h$ of $l$ points from $S \cap$ $\bar{B}(p; R)$ have the upper bound $\dfrac{2hR}{h+1}$ by Lemma B.5. If the $k$ smallest values of these sums are not greater than $\dfrac{2R}{h+1}$, which clearly holds for $h = 1$, these $k$ smallest values form the required row $a(h; 1) \leq \cdots \leq a(h; k)$ of the point $p = p_0$ in $\mathrm{PDD}^{\{h\}}(S; k)$. Indeed, in this case for any $h$ points $p_1, \ldots, p_h \in S$ with at least one distance (say) $|p_h - p_0| > R$, the lower bound of Lemma B.5 implies that the average sum $\dfrac{2}{h(h+1)} \sum_{0 \leq i < j \leq h} |p_i - p_j| > \dfrac{2R}{h+1}$ cannot be among the sought after $k$ smallest values.

If we could not find $k$ smallest sums not greater than $\dfrac{2R}{h+1}$, we extend the radius $R$ to $hR$. Similar to the above argument for the smaller radius $R$, the lower bound of Lemma B.5 guarantees than any average sum involving at least one point at a distance $|p_h - p_0| > hR$ is greater than $\dfrac{2hR}{h+1}$ and hence cannot be among $k \leq \begin{pmatrix} l \\ h \end{pmatrix}$ smallest sums that were already considered for the smaller ball $\bar{B}(p; R)$. Hence the larger ball $\bar{B}(p; hR)$ is guaranteed to contain the required $k$ smallest sums.

To cover necessary neighbors of all points $p$ from a motif $M = S \cap U$, we further increase the radius $hR$ by $0.5d$ and will use the earlier upper bound $R \leq c(S) \sqrt[n]{b(h, k) + 1} + d$. Let the ball $\bar{B}(p; hR + 0.5d)$ contain $l$ points of $S$ in addition to its center $p$. The upper bound $l + 1 \leq \left( \dfrac{hR + 1.5d}{c(S)} \right)^n$ from Lemma B.6 and the earlier upper bound $R \leq c(S) \sqrt[n]{b(h, k) + 1} + 1.5d$, we get

$$l \leq \left( \frac{hR + 1.5d}{c(S)} \right)^n \leq \left( h \sqrt[n]{b(h, k) + 1} + \frac{(h + 1.5)d}{c(S)} \right)^n.$$

Lemma B.9 simplifies the last bound to

$$l \leq (2h)^n (b(h, k) + 1) + \left( \frac{(2h + 3)d}{c(S)} \right)^n.$$

Substituting $c(S) = \sqrt[n]{\dfrac{\mathrm{vol}[U]}{mV_n}}$, we get $\dfrac{d}{c(S)} = V_n \nu(U)m$, where $\nu(U) = \dfrac{d}{\sqrt[n]{\mathrm{vol}[U]}}$ is called the *skewness* of the unit cell $U$. To find $l$ nearest neighbors of all $m$ points $p$ from the motif $M = S \cap U$, we gradually extend the cell $U$ in spherical layers by adding shifted copies of $U$ until we get the upper union from Lemma B.6:

$$U_+ = U_+(0; hc(S) \sqrt[n]{b(h, k) + 1} + 1.5d) \supset \bar{B}(0; hR + 0.5d).$$

If $h = 1$ then $b(1, k) = k + 1$ and $l = k$. The $k$ nearest neighbors of each of $m$ points $p \in M$ can be found by sorting the distances from $p$ to all other $l \leq 2^n(k + 2) + (5V_n \nu(U)m)^n$ points in $U_+ \cap S$. The total number of operations is at most $ml \log l$ as required.

Now consider only $h \geq 2$ and simplify the last bound:

$$l \leq (2h)^n (b(h, k) + 1) + (2h + 3)^n (V_n \nu(U)m)^n \leq$$

$$\leq (2h + 3)^n \Big( b(h, k) + (V_n \nu(U)m)^n \Big).$$

For each of $m$ points $p \in M$, we consider all $\begin{pmatrix} l \\ h \end{pmatrix} \leq \dfrac{l^h}{h!}$ average sums of pairwise distances between $p$ and any $h$ of $l$ points $p_1, \ldots, p_h \in U_+$. By Lemma B.9, the previous upper bound of $l$ gives the number of average sums

$$\begin{pmatrix} l \\ h \end{pmatrix} \leq \frac{l^h}{h!} \leq \frac{(2h + 3)^{hn}}{h!} 2^h \Big( b(h, k)^h + (V_n \nu(U)m)^{hn} \Big).$$

Since $b(h, k) = b + 1$, where $k \geq \begin{pmatrix} b \\ h \end{pmatrix} \geq \dfrac{(b-h)^h}{h!}$, Lemma B.9 gives the following upper bound.

$$b(h, k)^h = (b+1)^h \leq 2^h((b-h)^h + (h+1)^h) \leq$$

$$\leq 2^h(h!k + (h+1)^h), \text{ so } \frac{b(h,k)^h}{h!} < 2^h k + \frac{(2h+2)^h}{h!}.$$

Then the earlier upper bound is simplified to

$$\begin{pmatrix} l \\ h \end{pmatrix} \leq (2h+3)^{hn} 2^h \Big( 2^h k + \frac{(2h+2)^h}{h!} + \frac{(V_n \nu(U)m)^{hn}}{h!} \Big)$$

$$= \frac{(2h+3)^{hn} 2^h}{h!} \Big( 2^h h! k + (2h+2)^h + (V_n \nu(U)m)^{hn} \Big).$$

Estimate the first two terms inside the brackets as follows:

$$2^h h! k + (2h+2)^h \leq (2^h h! + (2h+2)^h)k \leq (2h+3)^h k.$$

The last inequality follows from the difference of powers:

$$2^h h! \leq (2h+3)^h - (2h+2)^h = \sum_{i=0}^{h-1} (2h+3)^i (2h+2)^{h-1-i}.$$

The right hand side is greater than

$$h(2h+2)^{h-1} = h2^{h-1}(h+1)^{h-1} \geq 2^h h!$$

because $(h+1)^{h-1} \geq 2(h-1)!$ for any $h \geq 2$. Then the total number of points in $U_+ \cap S$ has the upper bound

$$N \leq \frac{2^h}{h!}(2h+3)^{hn} \Big( (2h+3)^h k + (V_n \nu(U)m)^{hn} \Big).$$

For each of $m$ points $p \in M$, we find their $k$ smallest sums by sorting at most $N$ values. The total number of operations for computing $\mathrm{PDD}^{\{h\}}(S; k)$ is at most $mN \log N$. $\qquad \square$

Thank you for reading all the proofs!

