# OpenReview forum: "Towards continuous machine learning on periodic crystals by ultra-fast invariants"
_ICLR.cc/2025/Conference — Submitted to ICLR 2025_

### Official Review · Reviewer_jEka · 2024-11-03

**Soundness:** 3
**Presentation:** 3
**Contribution:** 2
**Rating:** 3
**Confidence:** 2

**Summary:**

This paper presents a novel approach to periodic crystal representation by addressing limitations in existing methods for distinguishing unique crystalline structures. The authors propose a new invariant for periodic point sets, which is designed to improve the identification of unique crystals. This invariant provides a systematic approach to eliminating near-duplicate entries in material databases.

**Strengths:**

The paper seems to offer a substantial contribution to evaluating novelty in crystalline materials, presenting a practical and (as far as i can tell) theoretically sound method for ensuring data quality in large-scale material databases. This seems to address an important challenge in materials science research.

**Weaknesses:**

The relevance to ICLR is not particularly clear, as the work does not introduce a new ML method or focus on representations of ML models in the typical sense. The authors should address the relevance for iclr or machine learning in general to improve understanding by the ml community. A less technical introduction would also benefit understanding and readability. Overall, the paper may be better suited for a more specialized audience or a theory-focused computer science conference.

**Questions:**

- Consider rephrasing or omitting the last sentence of the discussion, as it seems somewhat out of place.
- The paper is dense with technical details; consider moving some to the appendix (e.g. some examples) to allow more space for intuitive explanations and discussion.
- Invest more effort in improving the illustrations, particularly graphs. Figure 6’s text is small; consider adjusting the color scheme in Figure 1 to make it clearer and remove some of the elements to make it less crowded.
- The work may be suitable for a theory-focused ml conference, as it introduces a well-justified quality metric for material datasets rather than a new ML method.
- The language at times seems overly opinionated, with phrases like “notoriously hard problem” and “avoid such embarrassment.” These could be rephrased to maintain a more objective tone.

---

> ### Author Response · Authors · 2024-11-16
> **Thank you for the review**
>
> >This paper presents a novel approach to periodic crystal representation by addressing limitations in existing methods for distinguishing unique crystalline structures. The authors propose a new invariant for periodic point sets, which is designed to improve the identification of unique crystals. This invariant provides a systematic approach to eliminating near-duplicate entries in material databases.
>
> Thank you for the good summary.
>
> >The paper seems to offer a substantial contribution to evaluating novelty in crystalline materials, presenting a practical and (as far as i can tell) theoretically sound method for ensuring data quality in large-scale material databases. This seems to address an important challenge in materials science research.
>
> Thank you for correctly highlighting the significance of the theoretical and practical results.
>
> >The relevance to ICLR is not particularly clear, as the work does not introduce a new ML method or focus on representations of ML models in the typical sense.
>
> The work provides distance metric for evaluating the novelty of any simulated crystals generated by ML.
>
> >The authors should address the relevance for iclr or machine learning in general to improve understanding by the ml community.
>
> After the first paragraph of the introduction described the ongoing crisis of mistrust in materials generation, lines 31-32 highlighted the relevance to the ml community: "Machine learning can avoid such embarrassment by embracing a new continuous approach to data."
>
> >A less technical introduction would also benefit understanding and readability.
>
> Could you please specify the line numbers, where the text can be improved?
>
> >the paper may be better suited for a more specialized audience or a theory-focused computer science conference.
>
> Do you agree that the paper fits at least two topics listed at https://www.iclr.cc/Conferences/2025/CallForPapers
>
> unsupervised, self-supervised, semi-supervised, and supervised representation learning
>
> applications to physical sciences (physics, chemistry, biology, etc.)
>
> >Consider rephrasing or omitting the last sentence of the discussion, as it seems somewhat out of place.
>
> Ok, thank you for the helpful advice.
>
> >The paper is dense with technical details; consider moving some to the appendix (e.g. some examples) to allow more space for intuitive explanations and discussion.
>
> Examples usually help to explain formally defined concepts. What specific details would you advice to move the appendix?
>
> >Invest more effort in improving the illustrations, particularly graphs. Figure 6’s text is small;
>
> Ys, the font size in Figure  can be easily increased.
>
> >consider adjusting the color scheme in Figure 1 to make it clearer and remove some of the elements to make it less crowded.
>
> What specific elements in Figure 1 would you advise to remove?
>
> >The work may be suitable for a theory-focused ml conference, as it introduces a well-justified quality metric for material datasets rather than a new ML method.
>
> This well-justified quality metric is highly relevant to the ML for materials discovery
>
> >The language at times seems overly opinionated, with phrases like “notoriously hard problem”
>
> The hardness of Problem 1.1 is justified by the fact that all cell-based representations of materials are discontinuous under almost any perturbation and hence are not practical for real data including measurement noise and atomic vibrations.
>
> >and “avoid such embarrassment.” These could be rephrased to maintain a more objective tone.
>
> How would you advise to update the words “avoid such embarrassment” in the light of the public claims "the discovery of 2.2 million new crystals – equivalent to nearly 800 years’ worth of knowledge" at https://deepmind.google/discover/blog/millions-of-new-materials-discovered-with-deep-learning/, which were factually rebutted in the peer-review work at https://pubs.acs.org/doi/10.1021/acs.chemmater.4c00643?
>
> Would it be possible to justify how the overall score 3 (reject) follows from the other given scores:
> Soundness: 3: good
> Presentation: 3: good
> Contribution: 2: fair
>
> We are looking forward to receiving replies to the questions asked above.

---

> ### Comment · Reviewer_jEka · 2024-11-22
>
> > Could you please specify the line numbers, where the text can be improved?
>
> Just adding a few sentences (2 -3) with high-level summary of the main arguments after line number 23 would be helpful I think. This might open the paper to a broader audience.
>
> > Do you agree that the paper fits at least two topics listed at https://www.iclr.cc/Conferences/2025/CallForPapers
> > unsupervised, self-supervised, semi-supervised, and supervised representation learning
> > applications to physical sciences (physics, chemistry, biology, etc.)
>
> Yes. I would agree with application to physical sciences
>
> > What specific elements in Figure 1 would you advise to remove?
>
> - The green boxes could be removed.
> - it would be helpful to explain the color code (cyan and yellow)
> - Does the color code represent the same in left and right panel?
> - Extending the caption would be helpful (e.g. briefly refer to the elements shown in the right panel in the caption)

---

> > ### Author Response · Authors · 2024-11-22
> > **Thank you for the reply**
> >
> > >Just adding a few sentences (2 -3) with high-level summary of the main arguments after line number 23 would be helpful I think.
> >
> > Line 23 is right before the title of section 1: "Motivations for continuous invariants of periodic crystals".
> >
> > The abstract is considered a high-level summary. The first paragraph after line 23 introduces the motivations.
> >
> > "Real data such as periodic crystals often have ambiguous representations in the sense that experimental databases contain many substantially different entries encoding near-duplicate crystals with essentially the same properties Peplow (2023). Using descriptors or representations that discontinuously change under tiny perturbations of input data can lead to unjustified claims Kr¨amer (2021) and even to ‘paper mills’ Bimler (2022) reporting thousands of ‘new’ materials without proof. These public investigations Francis (2023) already led to hundreds of retracted papers Chawla (2022). Machine learning can avoid such embarrassment by embracing a new continuous approach to data."
> >
> > We would appreciate more specific details than "sentences (2 -3) with high-level summary of the main arguments after line number 23".
> >
> > >>Do you agree that the paper fits at least two topics listed at https://www.iclr.cc/Conferences/2025/CallForPapers unsupervised, self-supervised, semi-supervised, and supervised representation learning applications to physical sciences (physics, chemistry, biology, etc.)
> >
> > >Yes. I would agree with application to physical sciences
> >
> > Thank you for confirming that the paper fits the list of topics at https://www.iclr.cc/Conferences/2025/CallForPapers
> >
> > Do you also agree the development of invariants of point sets under isometry fits the first topic "unsupervised, self-supervised, semi-supervised, and supervised representation learning", at least the unsupervised part?
> >
> > In fact, the new higher-order invariants PDD^h are also well-defined for any finite set of unordered points, not only periodic.
> >
> > >The green boxes could be removed.
> >
> > Ok, though they were added to clearer separate the two parts of Figure 1.
> >
> > >it would be helpful to explain the color code (cyan and yellow)
> >
> > These two colors were used only for a better contrast different cases: 3 yellow boxes of the smaller size,
> >
> > >Does the color code represent the same in left and right panel?
> >
> > The background is shown in the same light green color. We can change other colors to make them all different.
> >
> > >Extending the caption would be helpful (e.g. briefly refer to the elements shown in the right panel in the caption)
> >
> > Ok, the current caption for the right-hand side image "Right: almost any perturbation of atoms can arbitrarily scale up a unit cell and break the symmetry" can be extended to add "discontinuously scaled up unit cells are shown in different colors".
> >
> > Were the above questions all reasons for the score 3 (reject)?

---

### Official Review · Reviewer_QApk · 2024-11-03

**Soundness:** 3
**Presentation:** 3
**Contribution:** 2
**Rating:** 5
**Confidence:** 4

**Summary:**

This paper addresses identifying near duplicate crystal, using invariants computed from point distance distributions PDD. Apparently the authors found a fast method for detecting and describing near duplicate crystals, tested on retrieval experiments with several crystal databases.

**Strengths:**

The development of invariants to rapidly identify near duplicate structures is interesting.

**Weaknesses:**

This paper seems to use a lot of language, assumptions references only clear to those studying crystal structures, and thus it is unclear if or how it links to the broader machine learning community. For example, "points" are discussed throughout the paper, it’s only in discussion where it is mentioned these points are in 3D space. There is no mention of other work using point distance distributions PDDs, e.g. distance transforms in classic computer vision, invariants such as the scale invariant feature transform SIFT used to identify near duplicate photos on a repeating pixel lattice.

No discussion of other invariants, e.g. 3D SIFT used to characterize and detecting near duplicates based on non-repeating structures in human brain images, which would seem to work quite well in the case of redundant crystal structures, using precisely the same KD search tree structure mentioned in this work, however appear unknown to the authors, e.g.
[a] Chauvin, Laurent, et al. "Neuroimage signature from salient keypoints is highly specific to individuals and shared by close relatives." NeuroImage 204 (2020): 116208.

**Questions:**

Why is there no mention of invariants from well-known work outside of work repeating crystal structures? Particularly given identical technologies used for retrieval, e.g. KD-tree search trees, using invariants derived from non-repeating structures (e.g. human body anatomy) which would appear to be a more difficult problem? If these are irrelevant, it would be good to know why.

---

> ### Author Response · Authors · 2024-11-16
> **Thank you for the review**
>
> >This paper addresses identifying near duplicate crystal, using invariants computed from point distance distributions PDD. Apparently the authors found a fast method for detecting and describing near duplicate crystals, tested on retrieval experiments with several crystal databases.
>
> Thank you for the high-level summary
>
> >Strengths: The development of invariants to rapidly identify near duplicate structures is interesting.
>
> Thank you for understanding that artificially produced near-duplicates fuel the ongoing crisis of mistrust in materials discovery, which cannot be recognized by any cell-based descriptors but can now be provably identified by generically complete and continuous invariants PDD^h.
>
> >This paper seems to use a lot of language, assumptions references only clear to those studying crystal structures, and thus it is unclear if or how it links to the broader machine learning community. For example, "points" are discussed throughout the paper, it’s only in discussion where it is mentioned these points are in 3D space.
>
> Because materials are physical objects in 3D space, they are naturally represented by points in 3D space, see the first phrase of the abstract: "Periodic point sets model all solid crystalline materials (crystals) whose atoms can be considered zero-sized points with or without atomic types."
>
> In fact, all definitions and main results hold in any dimension, not restricted to 3D space, so Euclidean space R^n explicitly appears in Definition 1.2, Problem 1.2, Definition 3.1, Theorems 4.4 and 4.5.
>
> >There is no mention of other work using point distance distributions PDDs, e.g. distance transforms in classic computer vision, invariants such as the scale invariant feature transform SIFT used to identify near duplicate photos on a repeating pixel lattice.
>
> The pointwise distance distributions PDDs are referenced in lines 151, 174, 183, 533, 535, and PDDs are fully introduced in Definition 2.3. We can add any recommended references, for example to SIFT. However, all these "finite-case" invariants are hard to extend from *finite* pixel grids to *infinite* periodic lattices with any (non-orthogonal) basis as shown in Fig 1 (left).
>
> >No discussion of other invariants, e.g. 3D SIFT used to characterize and detecting near duplicates based on non-repeating structures in human brain images, which would seem to work quite well in the case of redundant crystal structures, using precisely the same KD search tree structure mentioned in this work, however appear unknown to the authors, e.g. [a] Chauvin, Laurent, et al. "Neuroimage signature from salient keypoints is highly specific to individuals and shared by close relatives." NeuroImage 204 (2020): 116208.
>
> We are happy to include this reference. However, human brain is a bounded object, not an infinite periodic point set.
>
> >Why is there no mention of invariants from well-known work outside of work repeating crystal structures? Particularly given identical technologies used for retrieval, e.g. KD-tree search trees, using invariants derived from non-repeating structures (e.g. human body anatomy) which would appear to be a more difficult problem? If these are irrelevant, it would be good to know why.
>
> Invariants of finite sets were reviewed in lines 110-115 and 130-139, but we can include the extra recommended references.
>
> The caption of Figure 1 (left) explained the challenge: "Left: any periodic point set can be given by many pairs (cell, motif), see Definition 1.1. Finite subsets of the same lattice within boxes or balls of the same cut-off size can be vastly different. Right: almost any perturbation of atoms can arbitrarily scale up a unit cell and break the symmetry."
>
> Invariants of finite sets cannot be easily extended to infinite periodic sets with continuous guarantees as in Problem 1.2 (c,d), because any minimal (by volume or reduced) unit cell of a periodic set can arbitrarily scale up under almost any perturbation of points (motivated by atomic vibrations). The proposed hierarchy of higher-order PDD invariants is a  very special exception by using only (sums of) distances to neighbors (not vectors to or indices/types of neighbors) because all distances change continuously under perturbations.
>
> You are right that, after an invariant with a distance metric is computed, the search for nearest neighbors in the space of objects (crystals, not atomic centers) can be done by many classical algorithms such as kd-trees, so feel free to recommend any helpful references for this general search.
>
> We would be happy to answer more questions.

---

> > ### Comment · Reviewer_QApk · 2024-11-22
> >
> > The authors are apparently unaware of alternative 3D scale-invariant technology, and deflect suggestions of comparison.
> >
> > The SIFT "scale-invariant feature transform" is specifically invariant to scalings and perturbations of the type the author mentions. Specifically, SIFT specifically addresses problems with "the same cut-off size" and "perturbation of atoms can arbitrarily scale" mentioned.
> >
> > The authors state SIFT is irrelevant because it is used to describe bounded objects, and not infinite periodic point sets, however this only strengthens the argument to mention or compare against SIFT since bounded (and highly variable) objects in real image data would seem much more difficult and practical to model than infinitely repeating regular crystal structure.
> >
> > I agree "distances to neighbors" type invariants as proposed by the authors are of some interest, particularly in describing crystal structures and symmetries. I am not an expert on crystal structures, but am aware these have been studied intensely for the better part of a century in chemistry, and that the SIFT method was specifically designed to improve upon "distances to neighbors" models in realistic data contexts computer vision. There are many references describing SIFT invariants in continuous spaces, e.g. Morse theory, heat kernels.
> > Florack, Luc, and Arjan Kuijper. "The topological structure of scale-space images." Journal of Mathematical Imaging and Vision 12 (2000): 65-79.
> > Sun, Jian, Maks Ovsjanikov, and Leonidas Guibas. "A concise and provably informative multi‐scale signature based on heat diffusion." Computer graphics forum. Vol. 28. No. 5. Oxford, UK: Blackwell Publishing Ltd, 2009.

---

> > > ### Author Response · Authors · 2024-11-22
> > > **Thank you for the reply**
> > >
> > > >The authors state SIFT is irrelevant because it is used to describe bounded objects, and not infinite periodic point sets
> > >
> > > Do you agree that the word "irrelevant" appeared in the review, not in our response, see the quotes below.
> > >
> > > ">>If these are irrelevant, it would be good to know why.
> > >
> > > >Invariants of finite sets were reviewed in lines 110-115 and 130-139, but we can include the extra recommended references."
> > >
> > > >it is used to describe bounded objects, and not infinite periodic point sets
> > >
> > > Correct.
> > >
> > > > this only strengthens the argument to mention or compare against SIFT since bounded (and highly variable) objects in real image data would seem much more difficult and practical to model than infinitely repeating regular crystal structure.
> > >
> > > Sorry if this sounds disappointing but distinguishing near-duplicates in infinite periodic structures is a much harder problem than for finite bounded objects. May we ask if the arguments from the rebuttal below were read?
> > >
> > > "The caption of Figure 1 (left) explained the challenge: "Left: any periodic point set can be given by many pairs (cell, motif), see Definition 1.1. Finite subsets of the same lattice within boxes or balls of the same cut-off size can be vastly different. Right: almost any perturbation of atoms can arbitrarily scale up a unit cell and break the symmetry."
> > >
> > > Invariants of finite sets cannot be easily extended to infinite periodic sets with continuous guarantees as in Problem 1.2 (c,d), because any minimal (by volume or reduced) unit cell of a periodic set can arbitrarily scale up under almost any perturbation of points (motivated by atomic vibrations). The proposed hierarchy of higher-order PDD invariants is a very special exception by using only (sums of) distances to neighbors (not vectors to or indices/types of neighbors) because all distances change continuously under perturbations.
> > >
> > > You are right that, after an invariant with a distance metric is computed, the search for nearest neighbors in the space of objects (crystals, not atomic centers) can be done by many classical algorithms such as kd-trees, so feel free to recommend any helpful references for this general search."
> > >
> > > >I agree "distances to neighbors" type invariants as proposed by the authors are of some interest, particularly in describing crystal structures and symmetries. I am not an expert on crystal structures, but am aware these have been studied intensely for the better part of a century in chemistry
> > >
> > > Yes, despite all these studies, Problem 1.2 remained unresolved, which fuels the crisis of trust in artificially generated data.
> > >
> > > >many references describing SIFT invariants in continuous spaces, e.g. Morse theory, heat kernels. Florack, Luc, and Arjan Kuijper. "The topological structure of scale-space images." Journal of Mathematical Imaging and Vision 12 (2000): 65-79. Sun, Jian, Maks Ovsjanikov, and Leonidas Guibas. "A concise and provably informative multi‐scale signature based on heat diffusion." Computer graphics forum. Vol. 28. No. 5. Oxford, UK: Blackwell Publishing Ltd, 2009.
> > >
> > > Thank you for the recommended papers, which we are happy to cite.  We could not find neither the acronym SIFT nor the word "transform" in these papers.
> > >
> > > Do you agree that the first paper does not discuss distance metrics and computational compelxities?
> > >
> > > The main result of the second paper is stated for manifolds, not discrete sets of points, quoted below:
> > >
> > > "Theorem 1. If the eigenvalues of the Laplace-Beltrami operators of two compact manifolds M and N are not repeated, and T is a homeomorphism from M to N, then T is isometric if and only if k_t^M (x, x) =k_t^N(T(x),T(x)) for any x ∈ M and any t > 0."
> > >
> > > Do you agree that the condition "k_t^M (x, x) =k_t^N(T(x),T(x)) for any x ∈ M and any t > 0" requires checking infinitely many equalities? The same paper confirms that computing this infinite-size invariant is impractical:
> > >
> > > "one can take the family of functions {kt (x, ·)}t>0 parametrized by t as a signature for each point x ∈ M. Unfortunately, the complexity of this signature would be extremely high, since for a single point signature {k_t (x, ·)}t>0 is a function defined on R+ ×M, the product of temporal and spatial domains. Even worse, it would be difficult to compare the signatures of two different points. To make sense of the comparison, one would have to align two signature functions, say k_t (x, ·) and k_t (x′, ·), which would require a map from the neighborhood of x to that of x′."
> > >
> > > If you still think that SIFT is relevant to Problem 1.2 for periodic structures, we would appreciate more exact references to definitions and theorems.

---

> > > > ### Comment · Reviewer_QApk · 2024-11-22
> > > >
> > > > With regard to the authors last point,
> > > >
> > > > "one can take the family of functions {kt (x, ·)}t>0 parametrized by t as a signature for each point x ∈ M. Unfortunately, the complexity of this signature would be extremely high, since for a single point signature {k_t (x, ·)}t>0 is a function defined on R+ ×M, the product of temporal and spatial domains. Even worse, it would be difficult to compare the signatures of two different points. To make sense of the comparison, one would have to align two signature functions, say k_t (x, ·) and k_t (x′, ·), which would require a map from the neighborhood of x to that of x′."
> > > >
> > > > The SIFT method specifically aligns two signature functions, i.e. establishing multiple mapings from the neighborhood of x to that of x′, and compares invariant signature functions at different points to identify similarities. In could be used for practical 3D crystal data using data electron density images.

---

> > > > > ### Author Response · Authors · 2024-11-22
> > > > > **thank you for showing the expertise in crystal data**
> > > > >
> > > > > >The SIFT method specifically aligns two signature functions, i.e. establishing multiple mapings from the neighborhood of x to that of x′, and compares invariant signature functions at different points to identify similarities.
> > > > >
> > > > > Could we please clarify if the input for SIFT comparisons is a pair of continuous scalar functions on a bounded domain in (say) R^3?  Though this input substantially differs, we would still appreciate references to rigorous solutions of this analog of Problem 1.2 for continuous scalar functions.
> > > > >
> > > > > >In could be used for practical 3D crystal data using data electron density images.
> > > > >
> > > > > Our input for a 3D crystal structure is a collection of (atom) coordinates and unit cell parameters, much simpler than an electron density 3D image, which is a continuous scalar function, which is always given by its discretization to a fixed volumetric grid.
> > > > >
> > > > > Could we ask how SIFT handles periodic boundary conditions and discontinuities of unit cells illustrated in Figure 1?
> > > > >
> > > > > This discussion shows that ICLR reviewers know more physics and chemistry than even required for the paper.

---

### Official Review · Reviewer_DaqF · 2024-11-04

**Soundness:** 4
**Presentation:** 4
**Contribution:** 4
**Rating:** 5
**Confidence:** 2

**Summary:**

The work presents a novel representation for periodic crystals that allows for efficient and effective identification of de-facto duplicates in current crystal databases. In particular the representation is Lipschitz-continuous under noise, i.e., tiny variations of crystal structures, e.g. caused by measurement noise, only yield tiny distances in representation space. The representation allows the efficient identification of duplicates or (noisy) near-duplicates in databases by pairwise comparisons in databases with in total almost two million entries, thus revealing a large number of previously unknown duplicates as well as labelling errors (same structure with different chemical composition, which is infeasible) in the databases. Last but not least, the proposed approach is able to distinguish all known counter-examples, i.e. examples indistinguishable by some previous representations.

**Strengths:**

The work presents novel representations for periodic crystals that allows for efficient and effective identification of (near-)duplicates in existing databases. Thus the work does not only yield highly relevant practical findings, but also prevents "adversarial attacks" via claimed-new "discoveries" of materials which are however de facto already known and were only possible by exploiting deficiencies of previously proposed representations.

**Weaknesses:**

The authors proposes a procedurally constructed representation, and "machine learning" in their work is limited to pairwise distance computation. In this regard, having "machine lerning" in the title is somewhat misleading. The contribution may be well suited also to be placed in a more application-oriented publication venue (Science?). However, it is clear that a representation (albeit engineered) with the stated properties that, furthermore, allows very fast distance computation is also highly relevant for any kind of machine learning on crystal structures, as manifested by the fact that previous works the authors build upon have appeared at NeurIPS and CVPR.

Note, I revised my score from 8 to 5 after contemplating the other Reviews and discussing with CLkj; see comment further down for details

**Questions:**

-- In Problem 1.2, I assume (c) (2) should be symmetry, i.e., I guess d(b,c) is a typo and should be d(b,a)?

-- In 1.2. (d), lambda is not defined

-- p.2 l. 93f "gigantic loophole": It would be helpful to the reader if you could again cite the relevant reference (given earlier in the introduction)

---

> ### Author Response · Authors · 2024-11-16
> **Thank you for understanding the significance of the proved results**
>
> Sorry about the slow reply caused by other commitments early this week.
>
> >The work presents a novel representation for periodic crystals that allows for efficient and effective identification of de-facto duplicates in current crystal databases. The representation allows the efficient identification of duplicates or (noisy) near-duplicates in databases by pairwise comparisons in databases with in total almost two million entries, thus revealing a large number of previously unknown duplicates as well as labelling errors (same structure with different chemical composition, which is infeasible) in the databases.
>
> Thank for correctly summarizing the main results.
>
> >the work does not only yield highly relevant practical findings, but also prevents "adversarial attacks" via claimed-new "discoveries" of materials which are however de facto already known and were only possible by exploiting deficiencies of previously proposed representations.
>
> Yes, the main strengths are concisely highlighted.
>
> >The authors proposes a procedurally constructed representation, and "machine learning" in their work is limited to pairwise distance computation.
>
> Problem 1.2 advocates for a new continuous approach to machine learning on any data involving real (not discretized) variables. Because of experimental noise, any real structure (material, molecule, or another physical object) always has slightly different ground truth data, also due to thermal vibrations. Hence all experimental structures live i continuous spaces, which don't need to be cut into artificially labeled finitely many classes. In this real-world case, accuracies such as prevision, recall, F1-score etc, should be replaced by a continuous distance metric that quantifies differences between infinitely many real structures. Hence Problem 1.2 is a rigorous goal, which is stated for crystals but makes sense for future machine learning on real data in continuous spaces.
>
> >The contribution may be well suited also to be placed in a more application-oriented publication venue (Science?).
>
> The paper fits at least two topics listed at https://www.iclr.cc/Conferences/2025/CallForPapers
>
> unsupervised, self-supervised, semi-supervised, and supervised representation learning
>
> applications to physical sciences (physics, chemistry, biology, etc.)
>
> >it is clear that a representation (albeit engineered) with the stated properties that, furthermore, allows very fast distance computation is also highly relevant for any kind of machine learning on crystal structures, as manifested by the fact that previous works the authors build upon have appeared at NeurIPS and CVPR.
>
> You are right that the other top CS venues accepted the relevant work on invariants of finite and periodic point sets.
>
> >-- In Problem 1.2, I assume (c) (2) should be symmetry, i.e., I guess d(b,c) is a typo and should be d(b,a)?
>
> Yes, thank you for spotting this typo: the symmetry means that d(a,b)=d(b,a).
>
> >-- In 1.2. (d), lambda is not defined
>
> Yes, we should have said "there is a lambda such that if Q is obtained by perturbing every point of S up to epsilon, then d(I(S), I(Q)) <= lambda * epsilon."
>
> >-- p.2 l. 93f "gigantic loophole": It would be helpful to the reader if you could again cite the relevant reference (given earlier in the introduction)
>
> Yes, this loophole (allowing to claim a "new" materials by disguising a known one) had been exposed (but not resolved) by references Peplow (2023), Bimler (2022), Francis (2023), and Chawla (2022) from the first paragraph in the introduction.
>
> We would be happy to answer more questions.

---

> ### Comment · Reviewer_DaqF · 2024-11-22
>
> Given the other reviews and discussing with CLkj, while I maintain my assessment that  the work appears to be of great value, it does seem that ICLR is not the most spot-on publication venue after all.
>
> In my initial review, I had inferred from two cited closely related works published at NeurIPS and CVPR that the application topic at hand has a community within the general AI community; The two cited works are
>
> -- Widdowson and Kurlin, "Recognizing rigid patterns of unlabeled point clouds by complete and continuous isometry invariants with no false negatives and no false positives", CVPR 2023, and
>
> -- Widdowson and Kurlin, "Resolving the data ambiguity for periodic crystals", NeurIPS 2022.
>
> However, I am not an expert in the application topic and I do not know their community; Hence I am not in a position that would allow me to champion the work. Furthermore, given the other Reviews, it seems that the precedence given by the two papers listed above (which are notably also both from same two authors) might not be as strong as I thought. Consequently, I am joining the consensus that the work would be better-placed in a Journal, either for a very broad audience in Science, or very targeted in a Crystallography journal.
>
> I am joining this consensus however not without stressing once more that the work does appear to solve a highly important real-world problem with significant practical impact on research progress in Materials Science.

---

> > ### Author Response · Authors · 2024-11-22
> > **Thank you for the reply**
> >
> > >Given the other reviews and discussing with CLkj, while I maintain my assessment that the work appears to be of great value
> >
> > Thank for confirming the great value of the work.
> >
> > Could we ask who is CLkj because none of the reviewers has this acronym?
> >
> > >In my initial review, I had inferred from two cited closely related works published at NeurIPS and CVPR that the application topic at hand has a community within the general AI community; The two cited works are
> > -- Widdowson and Kurlin, "Recognizing rigid patterns of unlabeled point clouds by complete and continuous isometry invariants with no false negatives and no false positives", CVPR 2023, and
> > -- Widdowson and Kurlin, "Resolving the data ambiguity for periodic crystals", NeurIPS 2022.
> >
> > Thank you for underscoring that NeurIPS and CVPR published applications to atomic structures.
> >
> > >I am not an expert in the application topic and I do not know their community
> >
> > Why is the knowledge of "their community" is needed for a scientific review of rigorously proved Theorems 4.1, 4.3, 4.4, 4.5, even without considering important experiments of detecting thousands of near-duplicates in the major experimental databases?
> >
> > >given the other Reviews, it seems that the precedence given by the two papers listed above (which are notably also both from same two authors) might not be as strong as I thought.
> >
> > Did you mean that ICLR publishes by "the precedence", not through rigorous reviewing? Is a scientific argument that the new work advancing the results form the past two quoted papers should not be published because "the precedence ... might not be as strong"?
> >
> > > I am joining the consensus that the work would be better-placed in a Journal, either for a very broad audience in Science, or very targeted in a Crystallography journal.
> >
> > This forum was expected to be an open discussion, not a channel for communicating the unjustified "consensus" because scientific decisions should be based on facts and logic, not on counting votes without giving justified reasons.
> >
> > Because "Science, or very targeted in a Crystallography journal" were recommended, could we ask if you are an editor in these journals, authorized to write such an invitation. If not, why do you think that "a very broad audience in Science" or crystallographers can understand better understand the technical definitions and main Theorems 4.1, 4.3, 4.4, 4.5 than mathematicians and computer scientists at ICLR?
> >
> > >I am joining this consensus
> >
> > Did you mean that this decision was based on majority voting, not on scientific arguments?
> >
> > >stressing once more that the work does appear to solve a highly important real-world problem with significant practical impact on research progress in Materials Science.
> >
> > Thank you for confirming that "the work does appear to solve a highly important real-world problem with significant practical impact on research progress in Materials Science".
> >
> > > I revised my score from 8 to 5 after contemplating the other Reviews and discussing with CLkj
> >
> > Taking in account the comment above, what are the reasons for revising score 8 to 5?
> >
> > Communicating "the consensus" without any discussion with the authors contradicts https://www.iclr.cc/Conferences/2025/ReviewerGuide quoted below:
> >
> > "The discussion phase at ICLR is different from most conferences in the AI/ML community. During this phase, reviewers, authors and area chairs engage in asynchronous discussion and authors are allowed to revise their submissions to address concerns that arise. It is crucial that you are actively engaged during this phase."
> >
> > We hope that all reviewers and also "unexplained" CLkj understand that this case requires an ethical investigation because the ICLR was invented to change the unjustified model of publishing, quoted below from https://yann.lecun.com/ex/pamphlets/publishing-models.html
> >
> > "A major issue is that our current system, with its emphasis on highly-selective conferences, is highly biased against innovative ideas and favors incremental tweaks on well-established methods. Ideas that turn out to be highly influential are sometimes held up for months (if not years) in reviewing purgatory, particularly if they require several years to come to maturity (there are a few famous examples, mentioned). The friction in our publication system is slowing the progress of our field. It makes progress incremental. And it makes our conferences somewhat boring".

---

### Official Review · Reviewer_x6Eg · 2024-11-07

**Soundness:** 1
**Presentation:** 3
**Contribution:** 3
**Rating:** 5
**Confidence:** 2

**Summary:**

This is a new method for searching crystals. Multiple databases were used for evaluation.

The authors claim large improvements from years to hours.

The paper does make some large claims. For example, in the abstract there is a free standing claim saying
"The proposed Lipschitz continuity under noise is a new essential requirement for machine learning on any
data objects that have ambiguous representations and live in continuous spaces." This is for some types of
neural networks not for all algorithms. Correct? Please clarify by providing evidence or citations supporting the claim that this is an "essential requirement" for all ML on ambiguous data in continuous spaces. Alternatively, please clarify if this applies only to certain types of ML algorithms.

The paper is written for an extremely specialized and narrowly focused domain and not for the general
ML crowds found in ICLR.

The authors will need to add more explanations for general purpose ICLR audience. Otherwise this paper should be submitted to a Crystallography journal.

The main contribution is that the authors can  convert any PDD{h} into a fixed-size matrix,
which can be flattened into a vector for easy comparisons, while keeping the continuity and almost
all invariant data.

The authors also directly talk to the reviewers which is unexpected. For example, "We thank all reviewers for supporting scientific integrity, now guaranteed by the proposed invariants." or "Thank you for reading all the proofs!". Such claims are informal and not relevant for a publication in ICLR.

**Strengths:**

This works showcases large speedups in comparing crystalline structures.
The works extends Widdowson & Kurlin, 2022.
The paper shows and extensive amount of proofs and theorems.

**Weaknesses:**

The paper is not written for ML audience.
The paper is more relevant for Acta Crystallographica or similar journal.
The work is unnecessarily theoretical.  Giving the impression that there is not much novelty.

The title says "Towards continuous machine learning .. " but no ML example has been provided. Only similarity metrics.

**Questions:**

It is not clear what is the theoretical vs algorithmic contribution of the work. Please explain.

It is not clear what is the difference with the paper Widdowson & Kurlin, 2022. Please explain.

It is not clear what is the underlying representation that allows for the acceleration. Please provide a detailed explanation of the data structure you use, and how it enables faster computations compared to previous methods.

It is not clear what is the relationship with other existing machine learning methods. Please explain.
For example, can you compare your method to particular ML techniques used in materials science or crystal structure analysis. Can this method be integrated with graph neural networks or other ML models used for predicting crystal properties? Such as resistance to temperature changes?

---

> ### Author Response · Authors · 2024-11-16
> **Thank you for the review**
>
> >This is a new method for searching crystals. Multiple databases were used for evaluation. The authors claim large improvements from years to hours.
>
> Thank you for the concise summary.
>
> >a free standing claim saying "The proposed Lipschitz continuity under noise is a new essential requirement for machine learning on any data objects that have ambiguous representations and live in continuous spaces." This is for some types of neural networks not for all algorithms. Correct?
>
> This claim was not for neural networks but for any physical objects such as crystals or molecules whose atomic coordinates are real variables, not restricted to any pixel grid. Noise in measurements and thermal vibrations imply that any experimental structures of the same material (especially under different temperatures) will be at least slightly different. Hence such physical objects live in continuous spaces and requires.
>
> >Please clarify by providing evidence or citations supporting the claim that this is an "essential requirement" for all ML on ambiguous data in continuous spaces. Alternatively, please clarify if this applies only to certain types of ML algorithms.
>
> The basic reference for atomic vibrations is Richard Feynman's first lecture on physics: Atoms in motion.
>
> The continuity of machine learning outputs under perturbations of input is necessary for any gradient-based machine learning, which assumes even the stronger condition of smoothness. The evidence that smoothness requires continuity is standard calculus: https://en.wikipedia.org/wiki/Smoothness.
>
> >The paper is written for an extremely specialized and narrowly focused domain and not for the general ML crowds found in ICLR.
>
> The paper fits at least two topics listed at https://www.iclr.cc/Conferences/2025/CallForPapers
>
> unsupervised, self-supervised, semi-supervised, and supervised representation learning
>
> applications to physical sciences (physics, chemistry, biology, etc.)
>
> Taking into account the guidance above, would it be possible please to justify the claims about "an extremely specialized and narrowly focused domain" and also provide an official description of the "the general ML crowds found in ICLR"?
>
> >The authors will need to add more explanations for general purpose ICLR audience.
>
> Could you please specify the line numbers where more explanations can help? In fact, the presentation was evaluated as good with score 3.
>
> >The main contribution is that the authors can convert any PDD{h} into a fixed-size matrix, which can be flattened into a vector for easy comparisons, while keeping the continuity and almost all invariant data.
>
> Thank you for highlighting this result. The main contributions
>
> >The authors also directly talk to the reviewers which is unexpected. For example, "We thank all reviewers for supporting scientific integrity, now guaranteed by the proposed invariants." or "Thank you for reading all the proofs!". Such claims are informal and not relevant for a publication in ICLR.
>
> This is the first complaint about thanking reviewers. Ok, no thanks.
>
> >The work is unnecessarily theoretical.
>
> Why do you think that theoretical contributions are unacceptable at ICLR?
>
> >Giving the impression that there is not much novelty.
>
> Could you please justify this impression?
>
> >The title says "Towards continuous machine learning .. " but no ML example has been provided. Only similarity metrics.
>
> Not similarity metrics, but proper distance metrics satisfying all metric axioms and Lipschitz continuity.
>
> Lines 96-97 highlighted the importance of metric axioms for clustering: "The metric axioms are essential for recognizing S isometric to Q by d(I(S), I(Q)) = 0. If the third (triangle) axiom in 1.2(c) fails with any positive error, clustering may not be trustworthy Rass et al. (2022)."
>
> Is clustering considered within unsupervised learning, which is the very first official topic of ICLR?
>
> >It is not clear what is the theoretical vs algorithmic contribution of the work. Please explain.
>
> Lines 104-106: "The contributions to notoriously hard Problem 1.2 are (1) new higher-order invariants, which distinguished
> all known counter-examples to the completeness of past descriptors, and (2) the ultra-fast detection of (near-)duplicates in the world’s largest databases of experimental materials."
>
> >It is not clear what is the difference with the paper Widdowson & Kurlin, 2022. Please explain.
>
> Definition 3.1 substantially extends the invariant PDD in Definition 2.3 from the case h=1 to any higher order h>1. All Theorems 4.1, 4.3, 4.4, 4.5 are new and have not appeared in any past papers.
>
> We continue below because of the size limit.

---

> > ### Author Response · Authors · 2024-11-16
> > **Continuation of the response above**
> >
> > >It is not clear what is the underlying representation that allows for the acceleration. Please provide a detailed explanation of the data structure you use, and how it enables faster computations compared to previous methods.
> >
> > The first criterion for good science is correctness, not the speed, because there is no point in developing ultra-fast calculators that output 2x2=5. The main advantage over the previous methods is the proved Lipschitz continuity under perturbations of points.
> >
> > >It is not clear what is the relationship with other existing machine learning methods. Please explain. For example, can you compare your method to particular ML techniques used in materials science or crystal structure analysis.
> >
> > Yes, all cell-based representations of materials are discontinuous under almost any perturbation and hence are not practical for real data including measurement noise and atomic vibrations.
> >
> > >Can this method be integrated with graph neural networks or other ML models used for predicting crystal properties?
> >
> > Yes, similar to the previous PDD invariants, which were used for property predictions in the papers at
> > https://www.nature.com/articles/s41598-024-59938-z
> > https://link.springer.com/article/10.1007/s40192-024-00351-9
> >
> > >Such as resistance to temperature changes?
> >
> > Changing temperature usually changes crystal structures at least slightly and can arbitrarily scale up a primitive (minimal by volume or reduced cell). The Lipschitz continuity of the proposed invariants PDD^h under such perturbations is proved in Theorem 4.1.
> >
> > >Soundness: 1: poor
> >
> > Would it be possible to justify please the score 1 for soundness?
> >
> > We are looking forward to receiving replies to the questions asked above.

---

> > > ### Comment · Reviewer_x6Eg · 2024-11-22
> > >
> > > After you answer it is more clear to me that this is not the correct venue for this work.
> > >
> > > I am not saying that the work is not important. I am saying that you are stretching your claims considering this as relevant to ICLR.
> > > You are also assuming that the audience is already familiar with advanced topics in physics and chemistry. That earlier versions of the work were accepted to similar journals does not mean that every new version of this work should continue being accepted to the same venue. It does not work like that.
> > >
> > > I would recommend submitting this to Science Advances or similar high impact journal that has a larger readership from computational chemistry or physics.
> > >
> > > You claim that this fits under unsupervised, self-supervised, semi-supervised, and supervised representation learning but you have no applications or examples of those in your manuscript.

---

> > > > ### Author Response · Authors · 2024-11-22
> > > > **Thank you for the reply**
> > > >
> > > > >After you answer it is more clear to me that this is not the correct venue for this work.
> > > >
> > > > Reviewer jEka has already agreed that the paper fits the topic "applications to physical sciences (physics, chemistry, biology, etc.)" from https://www.iclr.cc/Conferences/2025/CallForPapers. Do you agree with this?
> > > >
> > > > If not, would it be possible to explain the definition of "the correct venue" that is used in ICLR reviewing?
> > > >
> > > > > I am saying that you are stretching your claims considering this as relevant to ICLR.
> > > >
> > > > Could you please provide the official guidance of "relevant to ICLR"?
> > > >
> > > > >You are also assuming that the audience is already familiar with advanced topics in physics and chemistry.
> > > >
> > > > What advanced topics were meant? The basics of atomic theory from school physics? All necessary definitions were given.
> > > >
> > > > >That earlier versions of the work were accepted to similar journals
> > > >
> > > > What earlier versions and journals were meant?
> > > >
> > > > >"does not mean that every new version of this work should continue being accepted to the same venue"
> > > >
> > > > Did you mean that the ICLR does not publish any improvements of earlier work that appeared in ICLR or elsewhere?
> > > >
> > > > >I would recommend submitting this to Science Advances or similar high impact journal
> > > >
> > > > Why do you think that the readers of Science Advances can better understand Theorems 4.1, 4.3, 4.4, 4.5?
> > > >
> > > > >You claim that this fits under unsupervised, self-supervised, semi-supervised, and supervised representation learning but you have no applications or examples of those in your manuscript.
> > > >
> > > > The new higher-order invariants PDD^h in Definition 3.1 are generically complete invariant representations of finite and periodic point sets. These invariant representations fit even the title of ICLR: International Conference on Learning Representations.
> > > >
> > > > >Soundness: 1: poor
> > > >
> > > > Would it be possible to justify please the score 1 for soundness?

---

### Author Response · Authors · 2024-11-22
**message from the area chair**

Dear Area Chair, thank you for contacting us (now we know your acronym CLkj) and for confirming that

"Theoretical contributions to machine learning and AI are acceptable at ICLR".

---

### Comment · Program_Chairs · 2024-11-26

The program committee has reviewed the discussion on this forum. All discussion about papers should focus only on the intellectual merit of the paper. We remind all participants to review and follow the code of conduct for the conference and professional standard. Further exchange breaching the norm of the professionalism will cause participants to be removed from the conference, such as desk rejection of the paper without further review.

---

### Author Response · Authors · 2024-11-26
**focus only on the intellectual merit**

Dear reviewers and area chair,

We hope you agree with the advice of the program chairs to "focus only on the intellectual merit", so we do not need to discuss again that the paper fits the topic of "applications to physical sciences (physics, chemistry, biology, etc.)" listed at https://www.iclr.cc/Conferences/2025/CallForPapers

In fact, the keywords "continuous machine learning" in the title are highly relevant to all machine learning because most real data are given by real-valued (not discrete) coordinates. Even if we forget about atomic structures, scans of macroscopic objects include positions of corners, salient points and other features with continuous real values, which are all noisy. Experiments on finite (often manually labeled) datasets of discretizations, e.g. pixel-based images, only approximate real ground-truth data in the infinitely continuous world.

If a machine learning algorithm was not proved to be stable under noise, its output can substantially change under a slight perturbation of input. Because all gradient-based optimization implicitly assume that target functions are smooth (differentiable at  least once), the continuity is a necessary requirement for any degree of smoothness, see a hierarchy of related concepts at https://en.wikipedia.org/wiki/Smoothness

That is why Problem 1.2 formalizing the continuity in the Lipschitz form rigorously states the future requirements for learning continuous (stable-under-noise) representations for all machine learning on real data with continuous values and hence is highly relevant to the ICLR.

We are happy to include the recommended references on SIFT features and other descriptors of finite data, though we are still waiting for exact quotes or specific references to any past results that formally prove their Lipschitz continuity needed for detecting near-duplicates.

We were a bit surprised by the reference " Chauvin, Laurent, et al. "Neuroimage signature from salient keypoints is highly specific to individuals and shared by close relatives." NeuroImage 204 (2020): 116208" on the topic of brain imaging, which seems a bit away from our work on periodic materials.

More generally, Figure 1 illustrated why the continuity in the periodic case is much harder than for any finite data. The key reason is the discontinuity of traditional representations based on a minimal (by volume) periodic cell (also called primitive or reduced). Any such cell can be doubled (tripled or scaled by any factor) and then almost any perturbation of a single point makes this extended cell again minimal.

This discontinuity derails all past attempts to build continuous descriptors of periodic sets by using finite subsets or bounded cells.  That is why new higher-order invariants in Definitions 3.1 and 3.4 are highly non-trivial, also justified by Theorems 4.1, 4.3, 4.4, 4.5.

The discontinuity of past invariants allowed anyone to easily disguise a known periodic structure (by a slight perturbation that significantly changes a traditional cell-based representation) and claim a "new material" without justification. Sections 5-6 demonstrated the scale of the resulting duplication in the world's largest databases of experimental materials, obtained in hours instead of years (estimated for past tools, which are all discontinuous anyway).

Because any experimental structure can be easily perturbed without changing its physical properties, all such databases should be first cleaned by continuous invariants to avoid machine learning on skewed and unrealistic data.

The new higher-order invariants PDD^h distinguished all (infinitely many) known counter-examples to the completeness of past invariants. The computations in Example 3,.3 and B.1 were actually done by hand, which shows how easy the new invariants can be computed.

Thank you again for focusing on the intellectual merits of the paper.

---

### Meta-Review · Area_Chair_CLkj · 2024-12-19

**Metareview:**

The paper proposes a new representation for periodic crystals and a way to identify duplicates. This is an important problem, with applications in medicine.

In its current form, the paper is unaccessible to the general machine learning audience, as found unanimously by all reviewers. Addressing this point will require a major revision which will require another round of reviews.

It is claimed in the title that the framework is useful for all of continuous machine learning, however the paper purely focuses on a very specific application on crystals. To better support these claims, other machine learning applications will make the paper stronger and more accessible to a general audience.

Due to the above reasons, I recommend to reject the paper. I encourage the authors to take the reviewers feedback into account for a revision.

**Additional Comments On Reviewer Discussion:**

All reviewers initially found the paper to be interesting but hard to understand for a machine learning audience not familiar with crystals. In extensive discussions with the authors and among themselves, their opinions remained the same and reviewers came to a consensus to reject the work.

---

### Decision · Program_Chairs · 2025-01-22

Reject